# A Reductions Approach to Risk-Sensitive Reinforcement Learning with Optimized Certainty Equivalents

**Kaiwen Wang** [1 2]  **Dawen Liang** [2]  **Nathan Kallus** [1 2]  **Wen Sun** [1]

## Abstract

We study risk-sensitive RL where the goal is learn a history-dependent policy that optimizes some risk measure of cumulative rewards. We consider a family of risks called the optimized certainty equivalents (OCE), which captures important risk measures such as conditional value-at-risk (CVaR), entropic risk and Markowitz's mean-variance. In this setting, we propose two meta-algorithms: one grounded in optimism and another based on policy gradients, both of which can leverage the broad suite of risk-neutral RL algorithms in an augmented Markov Decision Process (MDP). Via a reductions approach, we leverage theory for risk-neutral RL to establish novel OCE bounds in complex, rich-observation MDPs. For the optimism-based algorithm, we prove bounds that generalize prior results in CVaR RL and that provide the first risk-sensitive bounds for exogenous block MDPs. For the gradient-based algorithm, we establish both monotone improvement and global convergence guarantees under a discrete reward assumption. Finally, we empirically show that our algorithms learn the optimal history-dependent policy in a proof-of-concept MDP, where all Markovian policies provably fail.

## 1. Introduction

In reinforcement learning (RL), the classical objective by which we measure how good is a policy is the expected cumulative rewards along the trajectory (Sutton, 2004; Sutton and Barto, 2018). However, the mean reward of a choice (the choice of policy in RL being an example of such a choice) is a *risk-neutral* objective that is often observed to be inconsistent with human preferences, causing Allais Paradoxes (Allais, 1990; Kahneman and Tversky, 2013) and calling for

[1]Cornell Tech [2]Netflix Research. Correspondence to: Kaiwen Wang <`kaiwenw.github.io`>. Work done as Netflix intern.

*Proceedings of the 42nd International Conference on Machine Learning*, Vancouver, Canada. PMLR 267, 2025. Copyright 2025 by the author(s).

alternative *risk-sensitive* objectives (Yaari, 1987; Bowling et al., 2023). This is especially true in safety-critical settings (Artzner et al., 1999; Coronato et al., 2020; Wang et al., 2023b). In risk-sensitive RL (RSRL), the objective is some risk measure of cumulative rewards under a policy rather than the mean cumulative rewards (Howard and Matheson, 1972; Artzner et al., 1999).

In this paper, we propose a framework for solving RSRL with any static risk measure that can be expressed as an optimized certainty equivalent (OCE) (Ben-Tal and Teboulle, 2007). Specifically, for a utility function $u : \mathbb{R} \to [-\infty, \infty)$, the static OCE of a policy $\pi$ is defined as

$$\text{OCE}_u(\pi) := \max_{b \in \mathbb{R}} \{ b + \mathbb{E}_\pi [u(\sum_{h=1}^{H} r_h - b)] \}, \quad (1)$$

where the expectation is w.r.t. the random cumulative reward $\sum_{h=1}^{H} r_h$ under trajectories from $\pi$. The OCE captures several important risks for different choices of $u$. For example, with the hinge utility $u(t) = \min(t/\tau, 0)$, $\text{OCE}_u$ becomes CVaR at level $\tau \in (0, 1]$, which measures the mean outcome among the worst $\tau$-fraction of cases (Rockafellar and Uryasev, 2000). Moreover, OCE with the quadratic utility recovers Markowitz's mean-variance (Markowitz, 1952) and the exponential utility recovers entropic risk (Föllmer and Schied, 2011). We present a primer on OCE and more examples in App. B. We remark that another risk-sensitive objective is the iterated risk, which captures risk at each time step and is orthogonal to our static risk objective (as we explain in related works Sec. 1.1).

A key technical challenge in OCE RL is that optimal policies are in general non-Markovian (*i.e.*, history-dependent), which renders this more complex than risk-neutral RL where a Markovian optimal policy always exists. In prior works, a promising solution is to augment the MDP with a new state that tracks the cumulative rewards so far, which has been studied for CVaR and spectral risks (Bäuerle and Ott, 2011; Bäuerle and Glauner, 2021; Bastani et al., 2022). In this paper, we show that the augmented MDP (AugMDP) can be generalized with an OCE-based reward function to solve OCE RL with Markovian augmented policies (Sec. 2.1). We then propose two meta-algorithms that reduce OCE RL to risk-neutral RL by using the AugMDP as a bridge.

First, in Sec. 3, we propose a meta-algorithm based opti-

mistic RL oracles and we prove that its OCE regret can be bounded by the RL oracle's regret within the AugMDP up to a factor depending on $u$ (Thm. 3.2). We show that this approach can be instantiated with many RL oracles including UCB-VI for tabular MDPs (Azar et al., 2017) and Rep-UCB (Uehara et al., 2022) for low-rank MDPs, generalizing prior results in CVaR RL (Wang et al., 2023a; Zhao et al., 2024) to the OCE setting. While prior works in RSRL only considered model-based algorithms, we show that our approach is also amenable to the model-free optimistic RL oracle called GOLF (Jin et al., 2021). This establishes the first risk-sensitive RL bounds for exogenous block MDPs (Efroni et al., 2022; Xie et al., 2023).

Next, in Sec. 4, we propose a gradient-based meta-algorithm that uses policy gradient oracles and prove two types of guarantees: (1) convergence to the optimal OCE policy (Thm. 4.2), and (2) monotone improvement in a risk lower bound (Thm. 4.4), a novel property in RSRL theory. We instantiate our method with the natural policy gradient (NPG) oracle (Kakade, 2001) and prove these two guarantees under the standard conditions of policy optimization (Agarwal et al., 2021), plus a discrete reward assumption. Finally, our numerical simulation validates that our meta-algorithms can indeed learn the optimal OCE policy in a simple MDP where all Markovian policies (*e.g.*, Dabney et al., 2018; Lim and Malik, 2022) must have sub-optimal performance. This validates our thesis that history-dependent, non-Markovian policies are required for OCE optimality.

In sum, our main contributions are the following:

1. We propose an optimistic meta-algorithm for OCE RL that generalizes all prior works in CVaR to the more general OCE setting. We also apply GOLF as a novel oracle and prove the first risk-sensitive bounds in the challenging exogenous block MDP. (Sec. 3)

2. We propose a gradient-based meta-algorithm for OCE RL that enjoys *both* local improvement and global convergence. These are the first finite-sample bounds for gradient-based RSRL algorithms. (Sec. 4)

3. In a numerical simulation, we show that our meta-algorithms indeed optimally solve OCE RL in a proof-of-concept MDP where Markovian policies provably have sub-optimal performance. (Sec. 5)

### 1.1. Related Works

**From the theoretical side**, prior works with risk-sensitive regret or PAC bounds are based on optimistic, model-based oracles for CVaR or entropic risk. For CVaR, Bastani et al. (2022); Wang et al. (2023a) proposed UCB-based algorithms for tabular MDPs by using the AugMDP proposed by Bäuerle and Ott (2011). Zhao et al. (2024) then extended this approach to low-rank MDPs using the Rep-UCB oracle

(Uehara et al., 2022) in the AugMDP. For entropic risk, Fei et al. (2020; 2021) proposed UCB-based algorithms for tabular MDPs based on exponential Bellman equations. These prior works are limited to model-based optimistic oracles and largely do not extend beyond tabular MDPs, while our work provides a general framework for applying both model-free and model-based oracles to rich-observation OCE RL, including the challenging exogenous block MDP (Efroni et al., 2022). Moreover, these prior works are limited to CVaR or entropic risk, while our work provides a unifying framework for the broad class of OCE risk measures.

Chen et al. (2024) derives regret bounds for static Lipschitz risks under function approximation which is more related to our setting. However, their optimistic algorithm requires bounded eluder dimension or witness rank, which does not easily apply to the challenging exogenous block MDP setting. Indeed, the Bellman eluder dimension can grow with the size of the exogenous state space which is exponentially large or infinite (Xie et al., 2023). In contrast, we employ a coverability argument in the augmented MDP to obtain the first PAC bound for exogenous block MDP in risk-sensitive RL. Moreover, our work also studies other methods such as policy gradients, which are complementary to optimistic algorithms.

There are also several prior works on gradient-based RSRL algorithms that prove *local and asymptotic convergence* (Chow and Ghavamzadeh, 2014; Tamar et al., 2015a;b; Chow et al., 2018; Greenberg et al., 2022). These guarantees are not ideal since (1) any local optima can be much worse than the global optima and (2) asymptotics do not quantify the rate of convergence which could be very slow. In contrast, we fill this gap by proving non-asymptotic *finite-sample* bounds for the *global convergence* of gradient-based algorithms, by leveraging recent advances in risk-neutral policy gradient theory (Agarwal et al., 2021; Xiao, 2022). Finally, we also prove monotone improvement which is a novel property for RSRL to the best of our knowledge.

**From the empirical side**, a popular framework for RSRL is distributional RL (DistRL), which has demonstrated state-of-the-art results in both online (Bellemare et al., 2017; Dabney et al., 2018; Keramati et al., 2020; Ma et al., 2020) and offline RL (Urpí et al., 2021; Ma et al., 2021). At a high level, these algorithms learn the reward-to-go distribution and then greedily act to optimize the risk of the learned distribution at each step. Unfortunately, this can diverge for some MDPs since it is not consistent with any Bellman equations. To address this, Lim and Malik (2022) proposed to apply DistRL in an AugMDP, but their algorithm still relies on the *strong assumption* that the optimal policy is Markov, which is false in our counterexample (Sec. 5).

Another solution is Trajectory Q-Learning (TQL) (Zhou et al., 2023) which learns history-dependent policies and dis-

tributional $Q$-functions. While learning history-dependent policies ensures that TQL converges to the optimal RSRL policy, the learning process can be quite costly in sample complexity, computation and memory since it essentially *ignores the Markov property of MDPs*. Instead, our approach searches over the minimal sufficient policy set, the Markovian policies in the AugMDP, to optimally solve OCE RL while also maintaining statistical and computational efficiency. Finally, TQL requires DistRL to learn the reward-to-go distribution, while our framework supports both regression-based and distribution-based methods for learning the AugMDP value function.

**Iterated risk (*a.k.a.* dynamic risk).** Another risk-sensitive RL objective is the iterated risk $\rho(r_1 + \rho(r_2 + \rho(r_3 + \dots)))$, where $\rho$ is some risk measure. The iterated risk, *a.k.a.* dynamic risk, captures risk of the rewards-to-go at *each time step* and has been studied by many works (Ruszczyński, 2010; Du et al., 2023; Lam et al., 2023; Xu et al., 2023; Rigter et al., 2023; Liang and Luo, 2024). Dynamic risk also has close relations to distributionally robust MDPs, where an adversary can perturb the transition kernel at each time step (Wiesemann et al., 2013; Kumar et al., 2023; Bennett et al., 2024). The main benefit of iterated risk is that there exist Bellman equations and an optimal Markovian policy like in the classic RL setting. However, iterated risk may be overly risky in certain cases and overly conservative in other cases, and is also less interpretable due to the nested structure (Lim and Malik, 2022). In contrast, our static risk $\rho(r_1 + r_2 + r_3 + \dots)$ models the trajectory-level risk rather than at the step-level and is more interpretable. We highlight that static and dynamic risk are orthogonal objectives and the objective is ultimately the user's choice.

## 2. Preliminaries

We consider an MDP with state space $\mathcal{S}$, action space $\mathcal{A}$, horizon $H$, transition kernels $P_h : \mathcal{S} \times \mathcal{A} \to \Delta(\mathcal{S})$ and conditional reward distributions $R_h : \mathcal{S} \times \mathcal{A} \to \Delta([0,1])$, where $\Delta(S)$ denotes the set of distributions on the set $S$. A history-dependent policy $\pi = (\pi_h)_{h \in [H]}$ interacts with the MDP in the following way: start from an initial state $s_1$, then, for each step $h = 1, 2, \dots, H$, sample an action $a_h \sim \pi_h(s_h, \mathcal{H}_h)$, collect a reward $r_h \sim R_h(s_h, a_h)$, observe the next state $s_{h+1} \sim P_h(s_h, a_h)$. Here, $\mathcal{H}_h = \{(s_i, a_i, r_i)\}_{i \in [h-1]}$ denotes the history so far. We denote the cumulative reward distribution under $\pi$ as $Z(\pi) := \sum_{h=1}^{H} r_h$, where randomness arises from both the MDP and the policy $\pi$. We also assume that $Z(\pi) \in [0,1]$ almost surely for all policies (Jiang and Agarwal, 2018). The goal of OCE RL is to learn a history-dependent policy that maximizes the OCE objective:

$$\mathrm{OCE}_u^\star := \max_{\pi \in \Pi_{\mathsf{HD}}} \mathrm{OCE}_u(\pi), \qquad (2)$$

where $\Pi_{\mathsf{HD}}$ denotes the set of history-dependent policies.

We focus on the online setting, where at each round $k = 1, 2, \dots, K$, the learner outputs a (history-dependent) policy $\pi^k \in \Pi_{\mathsf{HD}}$. The learner's goal is to minimize the total sub-optimality across $K$ rounds:

$$\mathrm{Reg}_{\mathrm{OCE}}(K) := \sum_{k=1}^{K} \mathrm{OCE}_u^\star - \mathrm{OCE}_u(\pi^k).$$

Probably approximately correct (PAC) bounds are upper bounds on $\mathrm{Reg}_{\mathrm{OCE}}(K)$ that hold with high probability over the randomness of the learner. In particular, PAC bounds imply that at least one policy has high OCE, *i.e.*, $\min_{k \in [K]} \{\mathrm{OCE}_u^\star - \mathrm{OCE}_u(\pi^k)\} \leq \frac{1}{K} \mathrm{Reg}_{\mathrm{OCE}}(K)$. If the learner only executes policy $\pi^k$ at round $k$, then $\mathrm{Reg}_{\mathrm{OCE}}(K)$ is also called a *regret* bound.

### 2.1. Augmented MDP for OCE

The key challenge in RSRL with static risk is that the optimal policies are history-dependent, as there are no Bellman-like equations. A promising solution from prior works is to augment the MDP with a scalar state, which has been studied for CVaR and spectral risks (Bäuerle and Ott, 2011; Bäuerle and Glauner, 2021) – in this augmented state space, there does exist a Markovian optimal policy, bypassing the need to maintain history-dependent policies. In this section, we describe the augmented MDP (AugMDP) and extend it to the OCE setting, generalizing prior constructions.

The AugMDP has states $(s_h, b_h) \in \mathcal{S}_{\mathsf{aug}} := \mathcal{S} \times [-1, 1]$, where $s_h$ is the original MDP state and $b_h$ is a new scalar state called the budget, which intuitively tracks the cumulative rewards so far. The initial budget $b_1 \in [0, 1]$ is chosen by the learning algorithm and the budget transitions via $b_{h+1} = b_h - r_h$ where $r_h$ is reward at step $h$. The reward of the AugMDP is defined as $r_{\mathsf{aug}}^h(s, b, a) = 0$ for $h < H$ and $r_{\mathsf{aug}}^H(s, b, a) = \mathbb{E}_{r \sim R_H(s,a)}[u(r - b)]$, where $V_u^{\max} = \max_{c \in [-1, 1]} |u(c)|$ measures the scale of $r_{\mathsf{aug}}$. The augmented reward encodes the utility $u$ and is critical for the following optimality theorem.

**Theorem 2.1.** *There exists an initial budget $b_1^\star \in [0, 1]$ s.t. the optimal risk-neutral $\pi_{\mathsf{aug}}^\star$ in the AugMDP with initial budget $b_1^\star$ achieves optimal OCE in the original MDP.*

Since the optimal risk-neutral policy $\pi_{\mathsf{aug}}^\star$ is Markovian and deterministic, this theorem crucially mitigates the need to maintain history-dependent policies. We note that $(\pi_{\mathsf{aug}}^\star, b_1^\star)$ is a special history-dependent policy in the original MDP. Thus, Thm. 2.1 proves that the set of $(\pi_{\mathsf{aug}}, b_1)$, where $\pi_{\mathsf{aug}}$ is a Markovian augmented policy and $b_1$ is an initial budget, is a special class of history-dependent policies that is sufficient for OCE optimality. Moreover, a natural implementation of Thm. 2.1 is to solve risk-neutral RL in the AugMDP, which is the key idea behind our algorithms. We remark that our AugMDP construction generalizes the CVaR AugMDP from (Bäuerle and Ott, 2011) to any OCE risk.

For the AugMDP, we also define its quality function $Q_{\mathsf{aug}}^{\pi,h}(s_h, b_h, a_h) = \mathbb{E}_\pi[u(\sum_{t=h}^H r_t - b_h) \mid s_h, a_h, b_h]$ and value function $V_{\mathsf{aug}}^{\pi,h}(s_h, b_h) = \mathbb{E}_\pi[Q_{\mathsf{aug}}^{\pi,h}(s_h, b_h, a_h) \mid s_h, b_h]$. Also, let $V_{\mathsf{aug}}^{\star,h}(s_h, b_h) = \max_{\pi \in \Pi_{\mathsf{HD}}} V_{\mathsf{aug}}^{\pi,h}(s_h, b_h)$ be the optimal value function of the AugMDP.

*Proof of Thm. 2.1.* The main step is to write $\mathrm{OCE}_u^\star$ in terms of the optimal value of the AugMDP:

$$
\begin{aligned}
\mathrm{OCE}_u^\star & \overset{(i)}{=} \max_{\pi \in \Pi_{\mathsf{HD}}} \max_{b \in [0,1]} \{b + \mathbb{E}[u(Z(\pi) - b)]\} \\
& = \max_{b \in [0,1]} \{b + \max_{\pi \in \Pi_{\mathsf{HD}}} \mathbb{E}[u(Z(\pi) - b)]\} \\
& \overset{(ii)}{=} \max_{b \in [0,1]} \{b + \max_{\pi \in \Pi_{\mathsf{HD}}} V_{\mathsf{aug}}^{\pi,1}(s_1, b)\} \\
& \overset{(iii)}{=} \max_{b \in [0,1]} \{b + V_{\mathsf{aug}}^{\star,1}(s_1, b)\}. \quad (3)
\end{aligned}
$$

Step (i) uses OCE's definition in Eq. (1) and $Z(\pi) \in [0,1]$ w.p. 1, so the $\max_b$ can be taken over $[0,1]$. Step (ii) uses the definition of $b_h$ and $r_{\mathsf{aug}}$ to deduce that $\mathbb{E}_\pi[u(\sum_h r_h - b_1)] = \mathbb{E}_\pi[u(r_H - b_H) \mid b_1] = \mathbb{E}_\pi[r_{\mathsf{aug}}^H(s_H, b_H, a_H) \mid b_1] = V_{\mathsf{aug}}^{\pi,1}(s_1, b_1)$ for any $\pi$ and $b_1$. Step (iii) uses the classical RL result that the optimal risk-neutral policy $\pi_{\mathsf{aug}}^\star$ achieves the maximum value $V_{\mathsf{aug}}^\star$ over history-dependent policies (Puterman, 2014). Thus, letting $b_1^\star = \arg\max_{b \in [0,1]} \{b + V_{\mathsf{aug}}^{\star,1}(s_1, b)\}$, we have:

$$
\begin{aligned}
& \mathrm{OCE}_u(\pi_{\mathsf{aug}}^\star, b_1^\star) \\
& = \max_{b \in [0,1]} \{b + \mathbb{E}_{\pi_{\mathsf{aug}}^\star, b_1^\star}[u(Z(\pi_{\mathsf{aug}}^\star, b_1^\star) - b)]\} \\
& \geq b_1^\star + \mathbb{E}_{\pi_{\mathsf{aug}}^\star, b_1^\star}[u(Z(\pi_{\mathsf{aug}}^\star, b_1^\star) - b_1^\star)] \\
& = b_1^\star + V_{\mathsf{aug}}^{\star,1}(s_1, b_1^\star) = \mathrm{OCE}_u^\star.
\end{aligned}
$$

This proves that $(\pi_{\mathsf{aug}}^\star, b_1^\star)$ achieves the optimal $\mathrm{OCE}_u^\star$. $\square$

**Remark: the AugMDP is easy to simulate.** The transition function for the augmented state $b_h$ is simply $b_{h+1} = b_h - r_h$, which is *known and deterministic*. This ensures that simulating the AugMDP is *computationally efficient*. This is also important for proving bounds in the AugMDP. For example, model-based algorithms do not need to learn the AugMDP transitions (since it is known), so they generally inherent the same bounds as in the original MDP.

## 3. Meta-Algorithm with Optimism

In this section, we propose a meta-algorithm for OCE RL based on optimistic RL oracles. Our framework generalizes existing works in CVaR RL to more OCE risk measures and to more complex MDPs such as exogenous block MDPs. At a high level, the meta-algorithm implements the intuition from Thm. 2.1 and has two parts: (1) we apply an RL oracle in the AugMDP to learn $\pi_{\mathsf{aug}}^\star$, and (2) then select the right initial budget $b_1$. We start by formally defining optimistic oracles, which are RL algorithms that can explore using optimistic value functions:

**Definition 3.1** (Optimistic oracle). At each round $k = 1, 2, \ldots, K$, the oracle OPTALG first outputs an optimistic value function $\widehat{V}_{1,k}(\cdot)$, then sees an initial state $s_{1,k}$, and finally outputs a policy $\pi^k$ which is rolled-out from $s_{1,k}$ to collect data. Two conditions must be satisfied:

1. (Optimism) $\widehat{V}_{1,k}(s_1) \geq V_1^\star(s_1) - \varepsilon_k^{\mathrm{opt}}$ for all initial states $s_1$, where $\varepsilon_k^{\mathrm{opt}}$ are slack variables;

2. (Bounded Regret) $\sum_{k=1}^K (\widehat{V}_{1,k}(s_{1,k}) - V_1^{\pi^k}(s_{1,k})) \leq V^{\max} \mathrm{Reg}_{\mathrm{Opt}}(K)$, where $V^{\max}$ is the scale of cumulative rewards and $\mathrm{Reg}_{\mathrm{Opt}}(K)$ is bounds the total suboptimality gap across $K$ rounds. For this to be useful, $\mathrm{Reg}_{\mathrm{Opt}}(K)$ should be sublinear in $K$, *i.e.*, $\mathcal{O}(\sqrt{K})$.

We remark that the conditions of OPTALG can be satisfied in the Augmented MDP by modifying standard optimistic algorithms such as UCB-VI, Rep-UCB and GOLF. For model-based oracles (*e.g.*, UCB-VI, Rep-UCB), the augmented transition kernel for $b$ is known and deterministic and thus introduces no extra statistical or computational complexity for model learning in the augmented MDP. For model-free oracles (*e.g.*, GOLF), we need to be more careful and redefine important concepts such as completeness and coverability in the augmented MDP. We provide detailed algorithms and proofs for satisfying OPTALG in App. D.

Given an oracle OPTALG, our optimistic meta-algorithm (Alg. 1) proceeds as follows: at each round $k = 1, 2, \ldots, K$, we query OPTALG in the AugMDP and obtain an optimistic value function $\widehat{V}_{1,k}(\cdot)$. Then, we compute the initial budget by solving:

$$
\widehat{b}_k \leftarrow \arg\max_{b_1 \in [0,1]} \{b_1 + \widehat{V}_{1,k}(s_1, b_1)\}. \quad (4)
$$

Finally, we give the initial state $(s_1, \widehat{b}_k)$ to OPTALG, receive policy $\pi^k$, and proceed to the next round. We now state our main result for Alg. 1.

**Theorem 3.2.** *Assuming that OPTALG satisfies Def. 3.1, running Alg. 1 for $K$ rounds ensures that:*

$$
\sum_{k=1}^K \mathrm{OCE}_u^\star - \mathrm{OCE}_u(\pi^k, \widehat{b}_k) \leq V_u^{\max} \mathrm{Reg}_{\mathrm{Opt}}(K) + \sum_{k=1}^K \varepsilon_k^{\mathrm{opt}}.
$$

This theorem is a deterministic statement which bounds Alg. 1's OCE regret by the oracle's AugMDP regret, up to a scaling of $V_u^{\max}$. Here, $V_u^{\max}$ intuitively measures the statistical hardness of estimating $\mathrm{OCE}_u$ relative to $\mathbb{E}$, which has $V_{\mathbb{E}}^{\max} = 1$. For instance, $V_{\mathrm{CVaR}_\tau}^{\max} = \tau^{-1}$. Another intuition for $V_u^{\max}$ is the Lipschitz constant of the risk measure (Liang and Luo, 2024; Chen et al., 2024). Please see App. B for more examples. We now instantiate this result with three optimistic oracles from the RL literature.

**Algorithm 1** Meta-algorithm for optimistic oracles

1: **Input:** number of rounds $K$, optimistic oracle OPTALG satisfying Def. 3.1.
2: **for** round $k = 1, 2, \ldots, K$ **do**
3:     Query OPTALG in AugMDP for value func. $\widehat{V}_{1,k}(\cdot)$.
4:     Compute initial budget $\widehat{b}_k$ via Eq. (4).
5:     Give initial state $(s_1, \widehat{b}_k)$ to OPTALG and receive $\pi^k$.
6:     Collect a trajectory with $\pi^k$ starting from $(s_1, \widehat{b}_k)$.
7: **end for**

First, UCB-VI is a model-based algorithm for *tabular MDPs* with near-minimax-optimal dependence on $|\mathcal{S}|, |\mathcal{A}|, K$ (Azar et al., 2017). While the AugMDP is not tabular due to the real-valued augmented state $b_h$, the augmented state has a *known and deterministic* transition ($b_{h+1} = b_h - r_h$); hence, UCB-VI *only needs to learn the original MDP's transition model* when operating in the AugMDP. Indeed, by adapting the proof of Wang et al. (2023a) in Thm. D.1, we prove that UCB-VI satisfies Def. 3.1's conditions in the Aug-MDP with $\varepsilon_k^{\text{opt}} = 0$ and $\text{Reg}_{\text{Opt}}(K) \leq \widetilde{\mathcal{O}}(\sqrt{|\mathcal{S}||\mathcal{A}|HK})$. Combined with Thm. 3.2, this immediately implies an OCE regret bound of $\widetilde{\mathcal{O}}(V_u^{\max}\sqrt{SAHK})$, which is also near-minimax-optimal in $|\mathcal{S}|, |\mathcal{A}|, K$ for tabular MDPs. Moreover, in App. D.4, we obtain tight rates for CVaR using second-order oracles, showing our upper bound is tight.

The second oracle we discuss is Rep-UCB (Uehara et al., 2022), a model-based optimistic algorithm for *low-rank MDPs* (defined in Def. D.2), which are MDPs with a rank $d$ transition kernel and potentially infinite state space (Agarwal et al., 2020); low-rank MDPs capture tabular MDPs and linear MDPs. Again, the original MDP being low-rank does not necessarily imply that the augmented MDP is low-rank. Fortunately, as before, the augmented state $b_h$ has a known transition function and thus Rep-UCB only needs to learn the low-rank transitions of the original MDP, even when running in the AugMDP. Indeed Zhao et al. (2024) proved that Rep-UCB enjoys the same PAC bound in the Aug-MDP as in original low-rank MDP. We adapt their proof in Thm. D.4 and show that Rep-UCB satisfies Def. 3.1's conditions in the AugMDP with $\text{Reg}_{\text{Opt}}(K) \leq \widetilde{\mathcal{O}}(H^3|\mathcal{A}|d^2\sqrt{K})$ and $\sum_k \varepsilon_k^{\text{opt}} \leq \text{Reg}_{\text{Opt}}(K)$. Together with Thm. 3.2, we thus have an OCE PAC bound of $\widetilde{\mathcal{O}}(V_u^{\max}H^3|\mathcal{A}|d^2\sqrt{K})$, generalizing Zhao et al. (2024) to any OCE risk measure.

We have presented two examples with model-based oracles, whose bounds naturally lifts to the AugMDP since the augmented state $b_h$ has a known and deterministic transition. In general, model-based oracles enjoy the same bounds in the AugMDP as in the original MDP, making them convenient oracles for our meta-algorithm. Next, we discuss a model-free oracle to solve the challenging exogenous block MDP, where it is not tractable to learn the transition model.

### 3.1. Bounds for exogenous block MDPs

We now present a third optimistic oracle called GOLF (Jin et al., 2021) and prove the first risk-sensitive bound in the challenging exogenous block MDP (Ex-BMDP) problem, due to Efroni et al. (2022).

**Definition 3.3.** An Ex-BMDP has latent states $(z_h^{\text{en}}, z_h^{\text{ex}}) \in \mathcal{Z}_h^{\text{en}} \times \mathcal{Z}_h^{\text{ex}}$, where $\mathcal{Z}^{\text{en}}$ is the endogenous part which is tabular, and $\mathcal{Z}^{\text{ex}}$ is the exogenous part which is arbitrarily large. The endogenous latent transition conditions on the action $z_{h+1}^{\text{en}} \sim P_h^{\text{en}}(z_h^{\text{en}}, a_h)$, while the exogenous does not $z_{h+1}^{\text{ex}} \sim P_h^{\text{ex}}(z_h^{\text{ex}})$. The observation state is emitted from the latents $s_h \sim o_h(z_h^{\text{en}}, z_h^{\text{ex}})$ and, importantly, there exists a unique function $\phi_h^\star$ such that $\phi_h^\star(s_h) = (z_h^{\text{en}}, z_h^{\text{ex}})$ recovers the latents from the observation.

GOLF is a value-based (model-free) RL algorithm that achieves optimism by optimizing over a version space of $Q$-functions (Jin et al., 2021). The version space consists of candidate $Q$-functions from a function class $\mathcal{F} = (\mathcal{F}_h)_{h \in [H]}, \mathcal{F}_h \subset \mathcal{S}_{\text{aug}} \times \mathcal{A} \to \mathbb{R}$ whose Bellman backup is consistent with the data up to statistical noise. Value-based methods have shown promise in important applications such as LLM post-training (Mudgal et al., 2024; Zhou et al., 2025; Wang et al., 2025b). Due to space constraints, we defer the formal definition of GOLF in the AugMDP to App. D.3.

The standard condition required for Bellman-consistent algorithms like GOLF is completeness, which is required for Bellman backups to be well-behaved (Tsitsiklis and Van Roy, 1996; Munos and Szepesvári, 2008). Since we apply GOLF in the AugMDP, we posit completeness w.r.t. the AugMDP Bellman operator denoted as $\mathcal{T}_{\text{aug}}^{\star,h}$.

**Assumption 3.4.** $\mathcal{T}_{\text{aug}}^{\star,h} f_{h+1} \in \mathcal{F}_h$ for all $f_{h+1} \in \mathcal{F}_{h+1}$.

Another condition we require is discrete rewards (Assump. 3.5), which we use for bounding coverability in the AugMDP (Xie et al., 2023). Many safety-critical RL problems have discrete rewards, *e.g.*, in sepsis treatment, reward is the patient's discharge status (Johnson et al., 2016).

**Assumption 3.5.** We have a finite set $\mathcal{B} \subset [0, 1]$ such that $Z(\pi) \in \mathcal{B}$ w.p. 1 for all $\pi$, where $Z(\pi)$ is the cumulative reward of roll-outs from $\pi$.

This assumption can be satisfied by discretizing rewards as is common in distributional RL (Bellemare et al., 2017; Imani et al., 2024; Ayoub et al., 2024; Wang et al., 2025a) and in theory (Bastani et al., 2022; Wang et al., 2023a).

We now state our main result for Ex-BMDPs.

**Theorem 3.6.** *For $\delta \in (0, 1)$, under Assumps. 3.4 and 3.5, running Alg. 1 with OPTALG as GOLF (Alg. 5 in App. D.3) ensures that w.p. $1 - \delta$,*

$$\sum_{k=1}^{K} \text{OCE}_u^\star - \text{OCE}_u(\pi^k, \widehat{b}_k)$$
$$\leq \widetilde{\mathcal{O}}(V_u^{\max}H\sqrt{|\mathcal{B}||\mathcal{Z}^{\text{en}}||\mathcal{A}|K\log(|\mathcal{F}|/\delta)}).$$

Notably, the bound scales with the endogenous state size $|\mathcal{Z}^{\text{en}}|$ and the function class's complexity $\log(|\mathcal{F}|)$; indeed, all salient problem parameters matches the original risk-neutral bound from Xie et al. (2023). The completeness assumption may be weakened to realizability if we have local simulator access (Mhammedi et al., 2024); but otherwise is unavoidable even in risk-neutral RL (Jin et al., 2021; Xie et al., 2023). Thus, Thm. 3.6 demonstrates the versatility of our meta-algorithm to solve challenging MDPs that go beyond tabular and low-rank.

To prove Thm. 3.6, we need to show that GOLF satisfies the OPTALG conditions in Def. 3.1 with $\text{Reg}_{\text{Opt}}(K) \leq \widetilde{\mathcal{O}}(H\sqrt{|\mathcal{B}||\mathcal{Z}^{\text{en}}||\mathcal{A}|K})$ and $\varepsilon_k^{\text{opt}} = 0$. Due to completeness, we can invoke GOLF's coverability regret bound from Xie et al. (2023). Then, we prove that the AugMDP coverability is bounded by $|\mathcal{B}||\mathcal{Z}^{\text{en}}||\mathcal{A}|$ in Ex-BMDPs via a change-of-measure argument. Then, Thm. 3.6 immediately follows from Thm. 3.2. Please see App. D.3 for the full proof.

**Remark on Discrete Rewards.** It is possible to relax Assump. 3.5 by rounding rewards to fixed bins spaced $\epsilon$ apart. This discretization of width $\epsilon$ introduces an additional regret of at most $O(V_u^{\max} K \epsilon)$ and creates $|\mathcal{B}| = O(1/\epsilon)$ (Wang et al., 2023a). Thus, if an algorithm has a regret bound of $O(\sqrt{|\mathcal{B}|K})$, then applying this discretization would yield a regret bound of $O(V_u^{\max}\sqrt{K/\epsilon} + V_u^{\max} K \epsilon)$. Then, we can choose $\epsilon = \Theta(K^{-1/3})$ to yield a regret bound of $O(V_u^{\max} K^{2/3})$ which is sublinear. Thus, even when rewards are continuous, we can avoid Assump. 3.5 by discretizing rewards and paying a regret with rate $O(K^{2/3})$ instead of the minimax-optimal rate of $O(K^{1/2})$.

### 3.2. Proof for Main Reduction (Thm. 3.2)

*Proof of Thm. 3.2.* First, we upper bound $\text{OCE}_u^\star$:

$$\text{OCE}_u^\star \overset{(i)}{=} b_1^\star + V_{\text{aug}}^{\star,1}(s_1, b_1^\star)$$
$$\overset{(ii)}{\leq} b_1^\star + \widehat{V}_{1,k}(s_1, b_1^\star) + \varepsilon_k^{\text{opt}}$$
$$\overset{(iii)}{\leq} \widehat{b}_k + \widehat{V}_{1,k}(s_1, \widehat{b}_k) + \varepsilon_k^{\text{opt}},$$

where (i) is by Sec. 2.1, (ii) is by optimism of the oracle (first condition of Def. 3.1), and (iii) is by the definition of $\widehat{b}_k$ via Eq. (4). Second, we lower bound $\text{OCE}_u(\pi^k, \widehat{b}_k)$:

$$\text{OCE}_u(\pi^k, \widehat{b}_k) = \max_{b \in [0,1]}\{b + \mathbb{E}[u(Z(\pi^k, \widehat{b}_k) - b)]\}$$
$$\geq \widehat{b}_k + \mathbb{E}[u(Z(\pi^k, \widehat{b}_k) - \widehat{b}_k)]$$
$$= \widehat{b}_k + V_{\text{aug}}^{\pi^k,1}(s_1, \widehat{b}_k).$$

Thus, $\sum_k \text{OCE}_u^\star - \text{OCE}_u(\pi^k, \widehat{b}_k)$ is bounded by:

$$\leq \sum_k \widehat{V}_{1,k}(s_1, \widehat{b}_k) - V_{\text{aug}}^{\pi^k,1}(s_1, \widehat{b}_k) + \varepsilon_k^{\text{opt}}$$
$$\leq V_u^{\max} \text{Reg}_{\text{Opt}}(K) + \sum_k \varepsilon_k^{\text{opt}},$$

---

**Algorithm 2** Meta-algorithm for PO oracles

1: **Input:** number of rounds $K$, initial budget set $\mathcal{B}$ (Assump. 3.5), oracle POALG (Def. 4.1).
2: Let $d_1^{\text{aug}} = (\delta(s_1), \text{Unif}(\mathcal{B}))$ be the init. augmented state distribution. Initialize POALG with $d_1^{\text{aug}}$.
3: **for** round $k = 1, 2, \ldots, K$ **do**
4:     Obtain policy $\pi^k$ and value estimate $\widehat{V}_1^{\pi^k}$ from running POALG in the AugMDP.
5:     Compute the initial budget $\widehat{b}_k$ with Eq. (5).
6:     Give init. augmented state $(s_1, \widehat{b}_k)$ to POALG.
7: **end for**

---

where the last inequality is due to the oracle's regret bound (second condition of Def. 3.1). $\qquad\square$

**Computational Complexity of Alg. 1.** There are two sources of computational cost. The first cost is OPTALG: from the examples we discussed, UCB-VI and Rep-UCB are computationally and oracle efficient while GOLF is inefficient due to its version space optimism (Dann et al., 2018). So long as OPTALG is computationally efficient, then calls to OPTALG should also be efficient. The second cost is $\widehat{b}_k$: computing Eq. (4) exactly is difficult since the objective is non-convex. However, it can be efficiently approximated by projecting rewards to a discrete grid with spacing $\iota = \mathcal{O}(1/K)$, so each round's Eq. (4) can be done in $\mathcal{O}(K)$ time. Notably, the approximation error per round is at most $H\iota$ and across $K$ rounds is at most $\mathcal{O}(H\iota \cdot K) = \mathcal{O}(H)$, which is a lower order term (Wang et al., 2023a, App. H). Thus, provided that the oracle is efficient, Alg. 1 is both computationally and statistically efficient for OCE RL.

## 4. Meta-Algorithm with Policy Optimization

In this section, we propose a meta-algorithm that uses standard policy optimization (PO) oracles, which enjoy a new type of guarantee called *local improvement* that ensures the policy's performance monotonically improves at each round. First, we formalize the conditions needed for the risk-neutral PO oracle, which we remark is satisfied by gradient-based algorithms such as REINFORCE, NPG and PPO (Kakade, 2001; Agarwal et al., 2021; Grudzien et al., 2022).

**Definition 4.1** (PO Oracle). Let $d_1$ denote an exploratory initial state distribution. At each round $k = 1, 2, \ldots, K$, the oracle POALG outputs a policy $\pi^k$ and its value estimate $\widehat{V}_1^{\pi^k}(\cdot)$ such that $\mathbb{E}_{s_1 \sim d_1}|V_1^{\pi^k}(s_1) - \widehat{V}_1^{\pi^k}(s_1)| \leq \varepsilon_k^{\text{po}}$. Then, there are two conditions on $\{\pi^k\}_{k \in [K]}$'s performance:

1. (Approx. Improvement) $V_1^{\pi^{k+1}}(d_1) \geq V_1^{\pi^k}(d_1) - \varepsilon_k^{\text{po}}$, where $V(d_1) := \mathbb{E}_{s_1 \sim d_1} V(s_1)$ and $\varepsilon_k^{\text{po}}$ are slack vars;

2. (Global Convergence) $\sum_{k=1}^K (V_1^\star(d_1) - V_1^{\pi^k}(d_1)) \leq$

$V^{\max} \operatorname{Reg_{PO}}(K)$ where $\operatorname{Reg_{PO}}(K)$ bounds the total sub-optimality gap of POALG across $K$ rounds.

Since this oracle will be applied in the AugMDP, the $V^{\pi^k}$ term will be set as $V_{\mathsf{aug}}^{\pi^k}$. Moreover, we remark that the small estimation error condition $\mathbb{E}_{s_1 \sim d_1}|V_1^{\pi^k}(s_1) - \widehat{V}_1^{\pi^k}(s_1)| \leq \varepsilon_k^{\mathrm{po}}$ can be satisfied by via off-policy evaluation (OPE) by using the data collected in prior rounds (Munos and Szepesvári, 2008; Chang et al., 2022; Kallus and Uehara, 2020). Given access to simulator, we can of course also estimate the value by collecting online Monte Carlo roll-outs.

Given the oracle POALG, our meta-algorithm (Alg. 2) proceeds as follows: at each round $k = 1, 2, \ldots, K$, we query POALG with the initial budget distribution $\operatorname{Unif}(\mathcal{B})$, where $\mathcal{B}$ is the set of possible initial budgets that we assume to be discrete (Assump. 3.5). We require Assump. 3.5 because PO oracles, unlike optimistic ones, do not strategically explore and thus rely on an exploratory initial state distribution even in risk-neutral RL (Agarwal et al., 2021). Then, POALG returns a policy $\pi^k$ and an estimate of its value $\widehat{V}_1^{\pi^k}(\cdot)$, with which we compute the initial budget $\widehat{b}_k$ via

$$\widehat{b}_k \leftarrow \arg\max_{b_1 \in \mathcal{B}}\{b_1 + \widehat{V}_{1,k}(s_1, b_1)\}. \tag{5}$$

This is different from the optimistic version (Eq. (4)) since the maximization is restricted to $\mathcal{B}$. We now state our main results for Alg. 2, starting with global convergence.

**Theorem 4.2** (Global Convergence). *Under Assump. 3.5 and assuming* POALG *satisfies the global convergence criterion of Def. 4.1, running Alg. 2 ensures that:*

$$\sum_{k=1}^{K} \operatorname{OCE}_u^{\star} - \operatorname{OCE}_u(\pi^k, \widehat{b}_k) \\ \leq |\mathcal{B}|(V_u^{\max} \operatorname{Reg_{PO}}(K) + 2\sum_k \varepsilon_k^{\mathrm{po}}).$$

This theorem bounds Alg. 2's sub-optimality by the PO oracle's sub-optimality, up to a factor of $|\mathcal{B}|V_u^{\max}$, plus the value estimation errors. Thus, if POALG has small value estimation errors and converges to the optimal policy, then Alg. 2 converges to the optimal OCE policy. We highlight that Thm. 4.2 is a finite-sample PAC bound whereas prior guarantees for risk-sensitive policy gradients have been asymptotic (Tamar et al., 2015a;b).

Next, toward stating the local improvement result, we introduce the risk lower bound (RLB) $\operatorname{RLB}^{(k)}$ defined as:

$$\operatorname{RLB}(\pi) := \max_{b_1 \in \mathcal{B}}\{b_1 + V_{\mathsf{aug}}^{\pi,1}(s_1, b_1)\}. \tag{6}$$

Notice that the lower bound is tight at $\pi^{\star}$: $\operatorname{RLB}(\pi^{\star}) = \operatorname{OCE}_u^{\star}$, due to Eq. (3). In general, $\operatorname{RLB}$ approximately lower bounds the true OCE of $(\pi^k, \widehat{b}_k)$, as the following lemma proves.

**Lemma 4.3** (RLB). *The* $\operatorname{RLB}^{(k)}$ *approximately lower bounds the true OCE of $\pi^k$ with initial budget $\widehat{b}_k$:*

$$\operatorname{RLB}^{(k)} - \operatorname{OCE}_u(\pi^k, \widehat{b}_k) \leq 2|\mathcal{B}|\varepsilon_k^{\mathrm{po}}. \tag{7}$$

*Moreover, the lower bound is tight on average:*

$$\sum_k \operatorname{OCE}_u(\pi^k, \widehat{b}_k) - \operatorname{RLB}^{(k)} \leq |\mathcal{B}|V_u^{\max} \operatorname{Reg_{PO}}(K).$$

We now state the approximate improvement guarantee.

**Theorem 4.4** (Local Improvement). *Under Assump. 3.5 and assuming* POALG *satisfies the approximate improvement criterion of Def. 4.1, running Alg. 2 ensures that:*

$$\forall k \in [K] : \operatorname{RLB}^{(k+1)} \geq \operatorname{RLB}^{(k)} - |\mathcal{B}|\varepsilon_k^{\mathrm{po}}.$$

This theorem shows that Alg. 2's policies are approximately improving at each step, up to an error $|\mathcal{B}|\varepsilon_k^{\mathrm{po}}$. Hence, if the value estimation error from POALG is small, then the new policy's RLB cannot be much worse than the current. While local improvement is known for risk-neutral RL (Agarwal et al., 2021), to the best of our knowledge, Thm. 4.4 is the first analog in risk-sensitive RL. In sum, Thm. 4.2 and Thm. 4.4 provide complementary global and local guarantees for OCE RL, stated in a finite-sample manner.

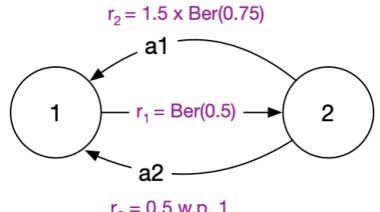

*Figure 1.* A simple MDP where the optimal CVaR policy is history-dependent. Each policy's cumulative reward distribution is shown in the table below.

| Action at $s_2$ | Cumulative-Reward Dist. | $\operatorname{CVaR}_{0.25}$ |
|---|---|---|
| $a_1$ (Markov) | $\{(0, \frac{1}{8}),\ (1, \frac{1}{8})$ $(1.5, \frac{3}{8}),\ (2.5, \frac{3}{8})\}$ | 0.5 |
| $a_2$ (Markov) | $\{(0.5, \frac{1}{2}),\ (1.5, \frac{1}{2})\}$ | 0.5 |
| $a_1$ if $r_1=0$ $a_2$ if $r_1=1$ | $\{(0, \frac{1}{8}),\ (1.5, \frac{7}{8})\}$ | 0.75 |

*Table 1.* The cumulative reward distribution of Markovian policies (rows 1-2) and the optimal policy (row 3). For the distribution, $\{(v_i, p_i)\}_i$ denotes a random variable that takes value $v_i$ w.p. $p_i$, s.t. $\sum_i p_i = 1$. The optimal CVaR policy has $\operatorname{CVaR}_{0.25}^{\star} = 0.75$ while both Markovian policies have $\operatorname{CVaR}_{0.25} = 0.5$.

### 4.1. Case Study: Natural Policy Gradient

As an example POALG, we consider natural policy gradients (NPG; Kakade, 2001) whose core ideas underpin TRPO and PPO (Schulman et al., 2015; 2017). Let $\pi_h^{\theta}$ denote

| OCE | Optimistic Alg (Alg. 1) | PO Alg (Alg. 2) | Best Markovian | Markovian Optimal? |
|---|---|---|---|---|
| $\mathbb{E}[X] - \mathrm{Var}(X)$ | $1.07 \pm 0.01$ | $1.06 \pm 0.01$ | 0.95 | ✗ |
| $\mathbb{E}[X] - 2\mathrm{Var}(X)$ | $0.81 \pm 0.01$ | $0.75 \pm 0.08$ | 0.5 | ✗ |
| $\mathrm{Entr}_{-1.0}(X)$ | $1.25 \pm 0.01$ | $1.23 \pm 0.03$ | 1.25 | ✓ |
| $\mathrm{Entr}_{-2.0}(X)$ | $0.90 \pm 0.02$ | $0.90 \pm 0.01$ | 0.91 | ✓ |
| $\mathrm{CVaR}_{0.25}(X)$ | $0.75 \pm 0.02$ | $0.71 \pm 0.08$ | 0.5 | ✗ |
| $\mathrm{CVaR}_{0.5}(X)$ | $1.12 \pm 0.03$ | $1.12 \pm 0.03$ | 1.0 | ✗ |

*Table 2.* We benchmark our optimistic Alg. 1 with UCB-VI as OPTALG and our PO Alg. 2 with NPG as POALG against the best Markovian policy for various OCEs. We repeat the experiment 10 times and report 95% confidence intervals for the average performance.

an augmented policy with parameters $\theta_h$. Then, the NPG update is given by

$$\theta_h^{k+1} = \theta_h^k + \eta F_{h,k}^{\dagger} \nabla_{\theta_h} V_{\mathsf{aug}}^{\pi^k}, \qquad (8)$$

where $F_{h,k} = \mathbb{E}_{\pi^k}[\nabla_\theta \log \pi_h^k(a_h \mid s_h, b_h) \nabla_\theta \log \pi_h^k(a_h \mid s_h, b_h)^\top]$ is the Fisher info matrix and $F_{h,k}^{\dagger}$ is its pseudo-inverse. A special class of policies are softmax policies: $\pi_h^\theta(a \mid s, b) \propto \exp(\theta_{s,b,a}^h)$ with $\theta_{s,b,a}^h \in \mathbb{R}$. Under softmax parameterization, the NPG update in Eq. (8) is equivalent to soft policy iteration (Kakade, 2001):

$$\pi^{k+1}(a \mid s, b) \propto \pi^k(a \mid s, b) \exp(\eta \cdot Q_{\mathsf{aug}}^{\pi^k}(s, b, a)).$$

We now show that NPG in the AugMDP satisfies both conditions in Def. 4.1. First, for local improvement, Agarwal et al. (2021, Lemma 5.2) implies that $\mathbb{E}_{b \sim \mathrm{Unif}(\mathcal{B})}[V_{\mathsf{aug}}^{\pi^{k+1}}(s_1, b) - V_{\mathsf{aug}}^{\pi^k}(s_1, b)] \geq 0$, which uses the fact that the initial budget distribution is $\mathrm{Unif}(\mathcal{B})$. Then, for global convergence, Agarwal et al. (2021, Theorem 5.3) proves that $\mathrm{Reg}_{\mathrm{PO}}(K) \leq \mathcal{O}(H)$, provided we set an appropriate learning rate $\eta = H \log |\mathcal{A}|$. Thus, NPG satisfies Def. 4.1 so we can apply both Thm. 4.2 and Thm. 4.4. To the best of our knowledge, these are the first non-asymptotic guarantees for gradient-based risk-sensitive RL. In App. E.1, we extend this analysis further to smooth policy parameterization and unknown $Q_{\mathsf{aug}}$ functions, using compatible function approximation à la Agarwal et al. (2021); Xiao (2022).

## 5. Simulation Experiments

We describe a numerical simulation to demonstrate the importance of learning history-dependent policies for OCE RL and to empirically evaluate our algorithms. Our code can be found at https://github.com/kaiwenw/oce-rl.

**Setting up synthetic MDP.** The proof-of-concept MDP is shown in Figure 1 and has two states. At $s_1$, all actions lead to a random reward $r_1 \sim \mathrm{Ber}(0.5)$ and transits to $s_2$. At $s_2$, the first action $a_1$ gives a random reward $r_2 \mid s_2, a_1 \sim 1.5 \cdot \mathrm{Ber}(0.75)$, while another action $a_2$ gives a deterministic reward $r_2 \mid s_2, a_2 = 0.5$. The trajectory ends after $s_2$.

The MDP is designed so that the optimal CVaR policy is non-Markovian. Due to the MDP's simplicity, we can compute the optimal CVaR and the Markovian policies' CVaR in closed form, which we list in Table 1. Specifically, the optimal action at $s_2$ depends on the random outcome of the first reward $r_1$: if $r_1 = 0$ then it should be risky and pick $a_1$; if $r_1 = 1$ then it should be conservative and pick $a_2$. This optimal policy achieves $\mathrm{CVaR}_{0.25}^\star = 0.75$, while Markovian policies, which do not react to $r_1$, can only achieve $\mathrm{CVaR}_{0.25} = 0.5$. This reinforces our thesis that history-dependent policies are required for optimality.

**Experiment with tabular policies.** We now apply our meta-algorithms to the synthetic MDP. For our optimistic meta-algorithm (Alg. 1), we use UCB-VI (Azar et al., 2017) as the oracle OPTALG. For our gradient-based meta-algorithm (Alg. 2), we use NPG as the oracle POALG. We evaluate our algorithms and compute the best Markovian policies for six OCEs: two CVaRs, two entropic risks and two mean-variances, where recall CVaR and entropic risks are defined as:

$$\mathrm{CVaR}_\tau(X) := \max_{b \in \mathbb{R}} \{b - \tau^{-1} \mathbb{E}[(b - X)_+]\},$$
$$\mathrm{Entr}_\beta(X) := \tfrac{1}{\beta} \ln \mathbb{E}[\exp(\beta X)].$$

The results are in Table 2 and we see that our algorithms are consistently better than the best Markovian policy. We remark that entropic risk is a special case where the optimal policy is Markovian (Fei et al., 2020; 2021); nonetheless, our algorithms still consistently learn the optimal policy despite AugMDP not being necessary for entropic risk.

**Experiment with neural network policies.** To test the versatility of our PO algorithm (Alg. 2), we also evaluate three deep RL oracles: PPO with forward KL (Schulman et al., 2017), PPO with backward KL (Hsu et al., 2020), and REINFORCE (Sutton et al., 1999). Using neural networks to approximate the policy and $Q$ functions, we train all three oracles to maximize $\mathrm{CVaR}_{0.25}$ in the synthetic MDP.

We plot the learning curves in Figure 2, where we see that Alg. 2 consistently converges to the optimal $\mathrm{CVaR}_{0.25}^\star$ value of 0.75 for all three oracles. An interesting trend is that

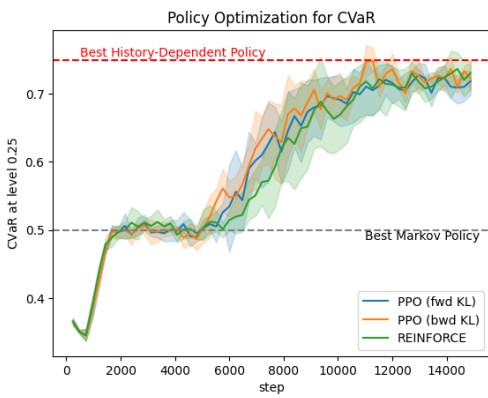

Figure 2. Learning curves for Alg. 2 with three oracles: REINFORCE and PPO with fwd & bwd KL. We repeat runs five times and report 95% confidence intervals for the mean performance.

Alg. 2 tends to first plateau at a Markovian policy at $\sim 2k$ steps, and then eventually converge to the optimal non-Markovian policy at $\sim 8k$ steps, suggesting that Markovian policies are perhaps easier-to-learn local optima in the AugMDP. Nonetheless, our algorithm consistently converges to the optimal policy given enough gradient steps. In sum, our experiments demonstrate the importance of history-dependent policies in risk-sensitive RL and the effectiveness of our AugMDP algorithms. We report all hyperparameters and training details in App. C.

## 6. Conclusion

In this paper, we proposed two meta-algorithms for RSRL with the static OCE risk. These meta-algorithms provide a general reduction from OCE RL to risk-neutral RL in the augmented MDP framework. First, we proposed an optimistic meta-algorithm (Alg. 1) that generalizes all prior bounds in CVaR RL to the OCE setting when UCB-VI and Rep-UCB are used as the optimistic oracle. Moreover, using GOLF as the optimistic oracle, we proved the first risk-sensitive regret bounds for exogenous block MDPs. Second, we proposed a gradient-based meta-algorithm (Alg. 2) that enjoys both global convergence and local improvement. With this framework, we deduced the first finite-sample bounds for policy gradients in risk-sensitive RL. Finally, we evaluate our algorithms in a proof-of-concept MDP where they consistently learn the optimal policy and outperform the best Markovian policy. A promising direction for future work is to scale our ideas to more complex tasks such as robotics or finetuning LLMs (Ouyang et al., 2022). Given the plurality of values and risks, it is also promising to apply our ideas to design algorithms for risk-sensitive pluralistic alignment (Sorensen et al., 2024; Wang et al., 2024a).

**Acknowledgements.** This work was supported by a Google PhD fellowship and grants NSF IIS-2154711, NSF CAREER 2339395, IIS-1846210, and DARPA LANCER: LeArning Network CybERagents.

## Impact Statement

This paper presents work whose goal is to advance the field of Machine Learning. There are many potential societal consequences of our work, none which we feel must be specifically highlighted here.

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

# Appendices

## A. List of Notations

*Table 3.* List of notations used in the paper.

| | |
|---|---|
| $\mathcal{S}, \mathcal{A}$ | State and action spaces. |
| $H$ | Time horizon. |
| $\Delta(S)$ | The set of distributions supported by set $S$. |
| $\Pi_{\mathsf{Markov}}$ | Class of Markovian policies that act only based on current state. |
| $\Pi_{\mathsf{HD}}$ | Class of history-dependent policies. |
| $\Pi_{\mathsf{aug}}, \Pi_{\mathsf{aug}}^{\mathsf{HD}}$ | Class of Markovian and history-dependent policies in the AugMDP. |
| $Q_{\mathsf{aug}}^{\pi,h}, V_{\mathsf{aug}}^{\pi,h}$ | $Q$ and value functions of $\pi$ in the AugMDP. |
| $Q_{\mathsf{aug}}^{\star,h}, V_{\mathsf{aug}}^{\star,h}$ | Optimal $Q$ and value functions in the AugMDP. |
| $(\pi, b)$ | Run the policy $\pi$ from an initial state $(s_1, b)$ in the AugMDP. |
| $Z(\pi)$ | The cumulative reward distribution of policy $\pi$. |
| $\mathrm{OCE}_u(\pi)$ | Optimized certainty equivalent with utility $u$ for $Z(\pi)$ (see App. B for primer on OCE). |
| $\mathrm{OCE}_u^\star$ | Optimal OCE of cumulative rewards by history-dependent policy, *i.e.*, $\max_{\pi \in \Pi_{\mathsf{HD}}} \mathrm{OCE}_u(\pi)$. |
| $\mathrm{RLB}^{(k)}$ | The risk lower bound at round $k$ for policy optimization algorithms (defined in Eq. (6)). |
| $V_u^{\max}$ | Scale of utility $u$ defined as $V_u^{\max} := \max_{c \in [-1,1]} \lvert u(c) \rvert$. |

## B. Primer on Optimized Certainty Equivalents (OCE)

In section is a short primer on optimized certainty equivalents (OCE), which capture many important risk measures including conditional value-at-risk (CVaR) (Rockafellar and Uryasev, 2000), entropic risk (Föllmer and Schied, 2011) and Markowitz's mean-variance (Markowitz, 1952). For a utility function $u : \mathbb{R} \to [-\infty, \infty)$, the OCE of a random variable $X$ is defined as:

$$\mathrm{OCE}_u(X) := \max_{b \in \mathbb{R}} \{b + \mathbb{E}[u(X - b)]\}.$$

In the paper, we focused on the case where $X$ is the cumulative reward distribution of some policy $\pi$, and the goal is to learn the policy that maximizes the OCE. While the OCE is well-defined for any utility function, there are common regularity conditions proposed by Ben-Tal and Teboulle (2007) that ensure the OCE is well-behaved. These conditions are:

[R1] $u$ is proper meaning that its domain is non-empty, *i.e.*, $\mathrm{dom}(u) := \{t \in \mathbb{R} : u(t) > -\infty\} \neq \emptyset$;

[R2] $u$ is closed meaning that its hypograph $\mathrm{hyp}(u) := \{(t, r) \in \mathbb{R} \times \mathbb{R} : r \leq u(t)\}$ is a closed set;

[R3] $u$ is concave meaning that its hypograph is convex;

[R4] $u$ is non-decreasing;

[R5] $u(0) = 0$ and $1 \in \partial u(0)$, where $\partial u(x) = \{g : \forall z \in \mathbb{R}, g \leq (u(z) - u(x))/(z - x)\}$ is the subgradient.

Ben-Tal and Teboulle (2007, Theorem 2.1) showed that, if $u$ satisfies R1-R5, then $\mathrm{OCE}_u$ enjoys the following properties:

[P1] Translation invariance: $\mathrm{OCE}_u(X + c) = \mathrm{OCE}_u(X) + c$ for constant $c \in \mathbb{R}$;

[P2] Monotonicity: if $X(\omega) \leq Y(\omega), \forall \omega \in \Omega$, then $\mathrm{OCE}_u(X) \leq \mathrm{OCE}_u(Y)$;

[P3] Concavity: for any $X, Y \in L_\infty, \lambda \in [0, 1]$, we have $\mathrm{OCE}_u(\lambda X + (1 - \lambda)Y) \geq \lambda \mathrm{OCE}_u(X) + (1 - \lambda) \mathrm{OCE}_u(Y)$;

[P4] Consistency: $\mathrm{OCE}_u(0) = 0$.

Translation invariance (P1) and consistency (P4) are intuitive since a deterministic payoff of $c$ should have a value of $c$. Monotonicity (P2) states that if $X$ is dominated by $Y$, then the risk of $X$ should be no greater than the risk of $Y$. Concavity (P3) implies that $-\mathrm{OCE}_u$ is a convex risk measure. However, we remark that these properties do not imply that $\mathrm{OCE}_u$ is a coherent risk measure (Artzner et al., 1999), which would require sub-additivity and positive homogeneity. Indeed, convexity is a relaxation of sub-additivity and positive homogeneity, which are not always satisfied by $\mathrm{OCE}_u$ even if $u$ satisfies R1-R5. For example, the entropic risk is an OCE with an exponential utility function satisfying R1-R5, but it is not a coherent risk measure.

In Table 4 below, we provide important examples of OCE, which includes (1) expectation, (2) entropic risk, (3) CVaR and (4) Markowitz's mean-variance. Recall that CVaR and Entropic risk are defined by:

$$\mathrm{CVaR}_\tau(X) := \max_{b \in \mathbb{R}}\{b - \tau^{-1}\mathbb{E}[(b - X)_+]\}, \qquad \mathrm{Entr}_\beta(X) := \tfrac{1}{\beta}\log\mathbb{E}\exp(\beta X).$$

| Risk Name | Parameter | Utility function $u$ | $V_u^{\max}$ |
|---|---|---|---|
| Mean ($\mathbb{E}X$) | None | $u(t) = t$ | 1 |
| Cond. Value-at-Risk ($\mathrm{CVaR}_\tau(X)$) | $\tau \in (0, 1]$ | $u(t) = -\tau^{-1}(-t)_+$ | $\tau^{-1}$ |
| Entropic Risk ($\mathrm{Entr}_\beta(X)$) | $\beta \in (-\infty, 0)$ | $u(t) = \frac{1}{\beta}(\exp(\beta t) - 1)$ | $\frac{1}{|\beta|}(\exp(|\beta|) - 1)$ |
| Mean-Variance ($\mathbb{E}X - c \cdot \mathrm{Var}(X)$) | $c > 0$ | $u(t) = t - ct^2$ if $t \le 1/(2c)$ else $u(t) = 1/(4c)$ | $1 + c$ |
| Mean-CVaR$_\tau$ ($\kappa_1\mathbb{E}X + (1 - \kappa_1) \cdot \mathrm{CVaR}_\tau(X)$) | $\kappa_1 \in [0, 1]$ | $u(t) = \kappa_1(t)_+ - \kappa_2(-t)_+$, where $\kappa_2 = \tau^{-1}(1 - \kappa_1) + \kappa_1$. | $\kappa_2$ |

*Table 4.* Examples of OCE risk measures with their corresponding $u$ and $V_u^{\max}$.

Table 4 also lists each OCE's $V_u^{\max}$ which recall is defined as $V_u^{\max} := \max_{c \in [-1,1]}|u(c)|$ and roughly measures the statistical hardness of learning $\mathrm{OCE}_u$ relative to $\mathbb{E}$ (which has $V_u^{\max} = 1$). We remark that OCE is a Lipschitz risk measures (Liang and Luo, 2024; Chen et al., 2024) with Lipschitz constant $V_u^{\max}$, and so $V_u^{\max}$ can also be interpreted in this way.

The first four rows of Table 4 are well-known risk measures and are known to be OCEs. The final row Mean-CVaR$_\tau$ is a lesser known example of OCE, but is perhaps more relevant in practice since it captures both the average and tail risks together. We now provide a proof that Mean-CVaR$_\tau$ is an OCE with the piecewise linear utility.

**Theorem B.1.** *Let $0 \le \kappa_1 < 1 < \kappa_2$ and consider the piece-wise linear utility:*

$$u_{\kappa_1,\kappa_2}(t) = \kappa_1(t)_+ - \kappa_2(-t)_+.$$

*where we denote $(y)_+ = \max(y, 0)$. Then, with $\tau = \frac{1-\kappa_1}{\kappa_2-\kappa_1}$, we have that*

$$\mathrm{OCE}_{u_{\kappa_1,\kappa_2}}(X) = \kappa_1\mathbb{E}X + (1 - \kappa_1)\mathrm{CVaR}_\tau(X).$$

*Proof.* By Ben-Tal and Teboulle (2007, Example 2.3),

$$\mathrm{OCE}_{u_{\kappa_1,\kappa_2}}(X) = \max_{b \in [0,1]}\{b + \kappa_1\mathbb{E}[(X - b)_+] - \kappa_2\mathbb{E}[(b - X)_+]\}. \tag{9}$$

Moreover, the optimal dual variable is $b^\star = F^\dagger((1 - \kappa_1)/(\kappa_2 - \kappa_1))$, where $F^\dagger(t) = \inf\{x \mid F(x) \ge t\}$ is the quantile function and $F$ is the CDF of $X$. Expanding Eq. (9) and using the fact that $(t)_+ - (-t)_+ = t$, we get

$$\mathrm{OCE}_{u_{\kappa_1,\kappa_2}}(X) = \kappa_1\mathbb{E}X + \max_{b \in [0,1]}\{(1 - \kappa_1)b - (\kappa_2 - \kappa_1)\mathbb{E}[(b - X)_+]\}$$

$$= \kappa_1\mathbb{E}X + (1 - \kappa_1)\max_{b \in [0,1]}\{b - \frac{\kappa_2 - \kappa_1}{1 - \kappa_1}\mathbb{E}[(b - X)_+]\}$$

$$= \kappa_1\mathbb{E}X + (1 - \kappa_1)\mathrm{CVaR}_\tau(X),$$

where $\tau = \frac{1-\kappa_1}{\kappa_2-\kappa_1}$. This completes the proof. $\square$

# C. More Details on Experimental Setup

## C.1. Network Architecture and Input Features

We parameterize both the augmented $Q$-values and the softmax policy using multilayer perceptrons (MLPs) with two hidden layers of dimension $64$. For the policy network, we one-hot encode the budget $b_h$ before passing it into the network. This transformation does not introduce approximation error since rewards are discrete in the Markov Decision Process (MDP). We observed that when $b_h$ was inputted directly as a real number, the training of the policy was less stable and often diverged. In contrast, the $Q$-network demonstrated greater robustness to how $b_h$ is featurized, likely due to the augmented value function being Lipschitz and monotonic in $b_h$. Unlike the policy network, the $Q$-network receives $b_h$ as a direct input without encoding. Finally, since the task is finite-horizon, we incorporate the time step $h$ as part of the state representation, following Pardo et al. (2018).

## C.2. Regularization

Log-barrier regularization played a crucial role in stabilizing and preventing the optimization from getting stuck due to vanishing gradients. Indeed in softmax policies, the gradients tend to vanish as the policy becomes more deterministic. To mitigate this issue, we applied a regularization weight of $0.1$ which helped to prevent the policy from becoming too deterministic. We remark that policy gradients with log-barrier regularization also have theoretical guarantees Agarwal et al. (2021, Theorem 12), making this a valid POALG for our PO algorithm.

## C.3. Hyperparameter Settings

| Component | Value/Description |
|---|---|
| Policy Network | Softmax policy with MLP with two hidden layers of dimension $64$ |
| Value Network | MLP with two hidden layers of dimension $64$ |
| Budget Encoding (Policy) | One-hot |
| Budget Encoding (Q-network) | Direct input |
| Optimizer | Adam with $\beta_1 = 0.9, \beta_2 = 0.999$ |
| Batch Size | 256 |
| Learning Rate | $5 \times 10^{-3}$ |
| GAE $\lambda$ | 0.95 |
| PPO KL weight | 0.1 |
| Regularization Weight | 0.1 |

*Table 5.* Hyperparameter settings used in our experiments.

# D. Proofs for Optimistic Meta-Algorithm

In this section, we study three illustrative examples of optimistic oracles that satisfy Def. 3.1 in the AugMDP. In particular, all these oracles can be used with our meta-algorithm Alg. 1 to form an optimistic OCE algorithm with the regret / PAC bounds in Thm. 3.2. The three oracles we consider are UCB-VI (Azar et al., 2017), Rep-UCB (Uehara et al., 2022) and GOLF (Jin et al., 2021).

First, UCB-VI is a model-based algorithm based on optimistic value iteration with exploration bonuses and achieves minimax-optimal regret in tabular MDPs (Azar et al., 2017). Second, Rep-UCB is a model-based algorithm based on elliptical exploration bonuses and achieves PAC bounds in low-rank MDPs (Uehara et al., 2022). While the AugMDP may initially seem more complex, since it is no longer tabular or low-rank, the regret bounds of these algorithms actually easily transfers to the AugMDP when the underlying MDP (*i.e.*, the MDP being augmented) is tabular or low-rank. The intuition is because the AugMDP does not introduce any new unknowns to the model, as the augmented transition and reward functions are known and deterministic, as highlighted in Sec. 2.1. Hence, the model that needs to be learned is the same in the original MDP and the AugMDP, and the prior regret bounds naturally translate to the AugMDP, which we formalize in the following subsections. In particular, we will see that prior works such as (Wang et al., 2023a; Zhao et al., 2024) have already implicitly proven that UCB-VI and Rep-UCB satisfies Def. 3.1 in the CVaR AugMDP.

The third optimistic oracle we consider is GOLF, a model-free algorithm based on version space optimism and can achieve regret bounds in exogenous block MDPs (Xie et al., 2023). Unlike model-based algorithm, extending GOLF to the AugMDP requires more care since the value functions being learned also takes in the augmented state $b$ as input; that is, the value function class is more complex in the AugMDP. To handle this added complexity, we posit a discreteness assumption in Assump. 3.5. Under this premise, we prove the previous GOLF analysis based on coverability can be extended to the AugMDP. Consequently, we derive the first OCE regret bounds in exogenous block MDPs (Efroni et al., 2022).

Finally, we conclude this section by showing how second-order bounds for the oracle can lead to tight and optimal regret for $\mathrm{CVaR}_\tau$ in tabular MDPs, recovering the main result of (Wang et al., 2023a). In particular, we observe that the key idea of the complex argument of (Wang et al., 2023a) is simply access to an oracle with second-order regret. We believe this observation can lead to tighter RSRL bounds, especially since second-order bounds have recently been possible in much more general MDPs via distributional RL (Wang et al., 2024b; 2025a; 2024c).

---

**Algorithm 3** Optimistic Oracle: UCB-VI (Azar et al., 2017)

---

1: **Input:** Number of rounds $K$, failure probability $\delta$.
2: **for** round $k = 1, 2, \ldots, K$ **do**
3:     Compute counts and empirical transition estimate,

$$N_k(s, a, s') = \sum_{h=1}^{H} \sum_{i=1}^{k-1} \mathbb{I}\left[(s_{h,i}, a_{h,i}, s_{h+1,i}) = (s, a, s')\right],$$

$$N_k(s, a) = 1 \vee \sum_{s' \in \mathcal{S}} N_k(s, a, s'), \quad \widehat{P}_k(s' \mid s, a) = \frac{N_k(s, a, s')}{N_k(s, a)},$$

4:     For all $s \in \mathcal{S}, b \in [0, 1]$, set $\widehat{V}_{H+1,k}(s, b) = u(-b)$.
5:     **for** $h = H, H-1, \ldots, 1$ **do**
6:         For all $s, b, a$,

$$\widehat{Q}_{h,k}(s, b, a) = \widehat{P}_k(s, a)^\top \mathbb{E}_{r_h \sim R(s,a)}\left[\widehat{V}_{h+1,k}(\cdot, b - r_h)\right] + \sqrt{\frac{\log(HSAK/\delta)}{N_k(s, a)}},$$

$$\pi_h^k(s, b) = \arg\max_a \widehat{Q}_{h,k}(s, b, a), \quad \widehat{V}_{h,k}(s, b) = \min\left\{\widehat{Q}_{h,k}(s, b, \pi_h^k(s, b)), V^{\max}\right\}.$$

7:     **end for**
8:     Output optimistic value function $\widehat{V}_{1,k}(s_1, \cdot)$.
9:     Receive adversarial $\widehat{b}_k$.
10:    Collect $\{(s_{h,k}, a_{h,k}, r_{h,k})\}_{h \in [H]}$ by executing $\pi^k$ starting from $(s_1, \widehat{b}_k)$ in AugMDP.
11: **end for**

---

### D.1. Example 1: UCB-VI

UCB-VI (Azar et al., 2017) is a model-based algorithm for tabular MDPs based on value iteration with exploration bonuses. Alg. 3 formalizes the UCB-VI algorithm in the AugMDP. The following theorem recovers the results from Wang et al. (2023a, Theorem 5.2), who focused on the more restricted CVaR RL setting.

**Theorem D.1.** *For any $\delta \in (0, 1)$, running Alg. 3 enjoys the following w.p. $1 - \delta$:*

1. *(Optimism) $\widehat{V}_{1,k}(s_1, b_1) \geq V_{\mathsf{aug}}^{\star,1}(s_1, b_1)$ for all $k \in [K], b_1 \in [0, 1]$;*

2. *(Regret) The regret in the AugMDP is at most,*

$$\sum_{k=1}^{K} \widehat{V}_{1,k}(s_1, \widehat{b}_k) - V_{\mathsf{aug}}^{\pi^k, 1}(s_1, \widehat{b}_k) \leq \widetilde{\mathcal{O}}(V_u^{\max}(\sqrt{SAHK \log(1/\delta)} + S^2 AH)).$$

   *where $\widetilde{\mathcal{O}}(\cdot)$ ignores terms logarithmic in $S, A, H, K$.*

*Proof Sketch.* Thm. D.1 can be proved by following the same argument in (Wang et al., 2023a, Appendix G.3), except that the CVaR utility $\tau^{-1} \min(t, 0)$ is replaced by any generic OCE utility $u$. In particular, optimism is ensured by Lemma G.3 and the Equation Bon★. The regret bound is ensured by the proof of Theorem 5.2 on Page 35. Thus, by applying the argument in (Wang et al., 2023a), we can show that UCB-VI satisfies the oracle conditions in Def. 3.1. □

Therefore, Thm. D.1 shows that UCB-VI satisfies the oracle conditions Def. 3.1 in the AugMDP with $\varepsilon_k^{\mathrm{opt}} = 0$ and $\mathrm{Reg}_{\mathsf{Opt}}(K) \leq \widetilde{\mathcal{O}}(\sqrt{SAHK \log(1/\delta)} + S^2 AH)$.

---

**Algorithm 4** Optimistic Oracle: Rep-UCB (Uehara et al., 2022)

---

1: **Input:** No. episodes $K$, Model Classes $(\Phi, \Upsilon)$.
2: Initialize $\mathcal{D}_h^{(0)} = \widetilde{\mathcal{D}}_h^{(0)} = \emptyset$ for all $h \in [H]$.
3: **for** round $k = 1, 2, \ldots, K$ **do**
4:     Learn the model via maximum likelihood estimation (MLE): for all $h \in [H]$:

$$\phi_h^{(k)}, \mu_h^{(k)} := \arg\max_{\phi \in \Phi, \mu \in \Upsilon} \sum_{s,a,s' \in \mathcal{D}_h^{(k-1)} \cup \widetilde{\mathcal{D}}_h^{(k-1)}} \log \phi(s_{h,i}, a_{h,i})^\top \mu(s_{h+1}, i)$$

5:     Define the empirical covariance $\Sigma_{h,k} = \sum_{s,a \in \mathcal{D}_h^{(k-1)}} \phi_h^{(k)}(s,a)(\phi_h^{(k)}(s,a))^\top + \lambda_k I$.
6:     Define the bonus, $b_{h,k}(s,a) = \min(\alpha_k \|\phi_h^{(k)}(s,a)\|_{\Sigma_{h,k}^{-1}}, 2)$ if $h < H$. Otherwise $b_{H,k} = 0$.
7:     Let $\widehat{Q}_{h,k}(s,b,a)$ denote quality function of the AugMDP with reward $r_{\mathsf{aug}}^{h,k}(s,b,a) + b_{h,k}(s,a)$ and transitions $\mathcal{P}_h^{(k)}(s' \mid s,a) = \phi_h^{(k)}(s,a)^\top \mu_h^{(k)}(s')$. Let $\widehat{V}_{h,k}(s,b) := \max_a \widehat{Q}_{h,k}(s,b,a)$ and $\pi^k := \arg\max_{a \in \mathcal{A}} \widehat{Q}_{h,k}(s,b,a)$. This can be achieved via a planning oracle (Zhao et al., 2024).
8:     Output optimistic value function $\widehat{V}_{1,k}(s_1, \cdot)$.
9:     Receive adversarial $\widehat{b}_k$.
10:    For each $h \in [H]$, execute $\pi^k$ starting from $(s_1, \widehat{b}_k)$ in AugMDP until $s_h, b_h$ and take two uniform actions. That is, $a_h \sim \mathrm{Unif}(\mathcal{A})$, observe $s_{h+1} \sim P_h^\star(s_h, a_h)$, and take $a_{h+1} \sim \mathrm{Unif}(\mathcal{A})$, observe $s_{h+2} \sim P_{h+1}^\star(s_{h+1}, a_{h+1})$. Then, add to datasets: $\mathcal{D}_h^{(k)} \leftarrow \mathcal{D}_h^{(k-1)} \cup \{(s_h, a_h, s_{h+1})\}$ and $\widetilde{\mathcal{D}}_{h+1}^{(k)} \leftarrow \widetilde{\mathcal{D}}_{h+1}^{(k)} \cup \{(s_{h+1}, a_{h+1}, s_{h+2})\}$.
11: **end for**

---

### D.2. Example 2: Rep-UCB

Rep-UCB (Uehara et al., 2022) is a model-based algorithm for low-rank MDPs based on elliptical bonuses. We first recall the definition of low-rank MDP (Agarwal et al., 2020).

**Definition D.2** (Low-Rank MDP). An MDP is has rank $d$ if its transitions have a low-rank decomposition $P_h(s' \mid s,a) = \phi_h^\star(s,a)^\top \mu_h^\star(s')$ where $\phi_h^\star(s,a), \mu_h^\star(s') \in \mathbb{R}^d$ are unknown features that satisfy $\sup_{s,a} \|\phi_h^\star(s,a)\|_2 \leq 1$ and $\|\int g(s') \mathrm{d}\mu_h^\star(s')\| \leq \|g\|_\infty \sqrt{d}$ for all $g : \mathcal{S} \to \mathbb{R}$.

Rep-UCB utilizes a model class $(\Phi, \Upsilon)$ to learn the low-rank transition $\phi_h^\star(s,a)^\top \mu_h^\star(s')$.

**Assumption D.3** (Realizability). $\phi_h^\star \in \Phi$ and $\mu_h^\star \in \Upsilon$ for all $h \in [H]$.

Rep-UCB as an oracle in the AugMDP is presented in Alg. 4. Our theory recovers the results of Zhao et al. (2024), who focused on the more restricted CVaR setting.

**Theorem D.4.** *Under Def. D.2 and Assump. D.3, for any $\delta \in (0, 1)$, Alg. 4 enjoys the following w.p. $1 - \delta$:*

1. *(Optimism) For all $k \in [K]$ and $b_1$, we have $V_{\mathsf{aug}}^{\star,1}(s_1, b_1) - \widehat{V}_{1,k}(s_1, b_1) \leq \mathcal{O}(H\sqrt{AL/k})$ where $L = \log(|\Phi||\Upsilon|HK/\delta)$;*

2. *(Total sub-optimality gap)*

$$\sum_{k=1}^K \widehat{V}_{1,k}(s_1, \widehat{b}_k) - V_{\mathsf{aug}}^{\pi^k,1}(s_1, \widehat{b}_k) \leq \mathcal{O}(V_u^{\max}(H^3 A d^2 \sqrt{KL})).$$

*Proof Sketch.* Thm. D.4 can be proved by following the same argument in Zhao et al. (2024), except that the CVaR utility $\tau^{-1}\min(t, 0)$ is replaced by any generic OCE utility $u$. In particular, optimism is ensured by Lemma C.2. The total sub-optimality gap bound was proven in Lemma C.1, where any $\tau^{-1}(t - R)^+$ can be replaced by $u(R - t)$. Thus, by applying the argument in (Zhao et al., 2024), we can show that Rep-UCB satisfies the oracle conditions in Def. 3.1. ☐

Therefore, Thm. D.4 shows that Rep-UCB satisfies the oracle conditions Def. 3.1 in the AugMDP with $\mathrm{Reg}_{\mathrm{Opt}}(K) \leq \mathcal{O}(H^3 A d^2 \sqrt{KL})$ and $\sum_k \varepsilon_k^{\mathrm{opt}} \leq \mathcal{O}(H\sqrt{AKL})$.

---

**Algorithm 5** Optimistic Oracle: GOLF (Jin et al., 2021)

---

1: **Input:** No. rounds $K$, Function class $\mathcal{F} = \mathcal{F}_1 \times \cdots \times \mathcal{F}_H$, threshold $\beta$.
2: Initialize $\mathcal{D}_h^{(0)} = \emptyset$ for all $h$, and $\mathcal{F}^{(0)} = \mathcal{F}$.
3: **for** round $k = 1, 2, \ldots, K$ **do**
4:     For each $b_1 \in [0,1]$, define $\widehat{Q}_k(\cdot; b_1) := \arg\max_{f \in \mathcal{F}^{(k-1)}} \max_{a \in \mathcal{A}} f_1(s_1, b_1, a)$.
5:     Output optimistic value functions $\widehat{V}_k(s, b; b_1) = \max_{a \in \mathcal{A}} \widehat{Q}_k(s, b, a; b_1)$.
6:     Receive adversarial $\widehat{b}_k$.
7:     Define policy $\pi_h^k(a \mid s, b) := \arg\max_{a \in \mathcal{A}} \widehat{Q}_{h,k}(s, b, a; \widehat{b}_k)$.
8:     Execute $\pi^k$ from $\widehat{b}_k$ in the AugMDP and collect a trajectory $(s_h^{(k)}, b_h^{(k)}, a_h^{(k)}, r_h^{(k)})_{h \in [H], k \in [K]}$.
9:     Update dataset $\mathcal{D}_h^{(k)} = \mathcal{D}_h^{(k-1)} \cup \{(s_h^{(k)}, b_h^{(k)}, a_h^{(k)}, r_h^{(k)}, s_{h+1}^{(k)}, b_{h+1}^{(k)})\}$.
10:    Compute confidence set:

$$\mathcal{F}^{(k)} \leftarrow \left\{ f \in \mathcal{F} : \mathcal{L}_h^{(k)}(f_h, f_{h+1}) \leq \min_{g \in \mathcal{F}_h} \mathcal{L}_h^{(k)}(g, f_{h+1}) + \beta, \forall h \in [H] \right\},$$

$$\text{where } \mathcal{L}_h^{(k)}(g, f') = \sum_{i \leq k} (g(s_h^{(i)}, b_h^{(i)}, a_h^{(i)}) - r_h^{(i)} - \max_{a \in \mathcal{A}} f'(s_{h+1}^{(i)}, b_{h+1}^{(i)}, a))^2$$

11: **end for**

---

### D.3. Example 3: GOLF

GOLF (Jin et al., 2021) is a model-free algorithm that establishes optimism by optimizing over a version space of candidate $Q$-functions. In this section, we show that GOLF's regret in the AugMDP can be bounded under a discreteness premise on the cumulative rewards (cf. Assump. 3.5).

We consider a function class $\mathcal{F} = \mathcal{F}_1 \times \cdots \times \mathcal{F}_H$ with elements $f = (f_1, \ldots, f_H) \in \mathcal{F}$ such that $f_h : \mathcal{S} \times [0,1] \times \mathcal{A} \to [-V_u^{\max}, V_u^{\max}]$. As convention, we set $f_{H+1}(s, b, a) = u(-b)$.

GOLF constructs the version space by selecting a subset of $\mathcal{F}$ such that the TD-error is small for all $h$. This is formalized in Alg. 5 with the squared TD-error. To ensure the regression at each step can succeed, GOLF requires Bellman completeness (Jin et al., 2021; Xie et al., 2023). To state Bellman completeness, we first define the Bellman optimality operator in the AugMDP: for a function $f : \mathcal{S} \times [0,1] \times \mathcal{A} \to \mathbb{R}$ and any $h < H$:

$$\mathcal{T}_{\mathsf{aug}}^{\star,h} f(s_h, b_h, a_h) = \mathbb{E}_{s_{h+1} \sim P_h(s_h, a_h), r_h \sim R_h(s_h, a_h)} [\max_{a' \in \mathcal{A}} f(s_{h+1}, b_h - r_h, a')],$$

$$\mathcal{T}_{\mathsf{aug}}^{\star,H} f_{H+1}(s_H, b_H, a_H) = \mathbb{E}_{r_H \sim R_H(s_H, a_H)} [u(r_H - b_H)].$$

**Assumption D.5** (AugMDP Bellman Completeness). $\mathcal{T}_{\mathsf{aug}}^{\star,h} f_{h+1} \in \mathcal{F}_h$ for all $f_{h+1} \in \mathcal{F}_{h+1}, h \in [H]$.

To prove regret bounds for exogenous block MDPs, we recall the definition of coverability from Xie et al. (2023). Coverability is a complexity measure that captures the minimum possible concentrability coefficient in the MDP:

$$\mathsf{Cov} := \min_{\mu_h \in \Delta(\mathcal{S} \times \mathcal{A})} \max_{\pi \in \Pi_{\mathsf{Markov}}, h \in [H]} \left\| \frac{d_h^\pi}{\mu_h} \right\|_\infty,$$

where $\Pi_{\mathsf{Markov}}$ is the set of policies in the original MDP. We extend this notion to the AugMDP:

$$\mathsf{Cov}_{\mathsf{aug}} := \min_{\mu_h \in \Delta(\mathcal{S} \times [0,1] \times \mathcal{A})} \max_{\pi \in \Pi_{\mathsf{aug}}, h \in [H], b_1 \in [0,1]} \left\| \frac{d_h^{\pi, b_1}}{\mu_h} \right\|_\infty.$$

We now state the AugMDP regret bound for GOLF.

**Theorem D.6.** *Under Assump. D.5, for any $\delta \in (0,1)$, running Alg. 5 with $\beta = \Theta(\log(KH|\mathcal{F}|/\delta))$ enjoys the following w.p. $1 - \delta$:*

    *1. (Optimism) $\widehat{V}_{1,k}(s_1, b_1; b_1) \geq V_{\mathsf{aug}}^{\star,1}(s_1, b_1)$ for all $k \in [K], b_1 \in [0,1]$;*

*2. (Regret bound)*

$$\sum_{k=1}^{K} \widehat{V}_{1,k}(s_1, \widehat{b}_k; \widehat{b}_k) - V_{\mathsf{aug}}^{\pi^k, 1}(s_1, \widehat{b}_k) \le \mathcal{O}\big(V_u^{\max} H \sqrt{\mathsf{Cov}_{\mathsf{aug}} \beta K \log(K)}\big).$$

*Proof Sketch.* Thm. D.6 is a direct consequence of Jin et al. (2021); Xie et al. (2023), since all that we have done is rewritten everything (algorithms, assumptions and theorems) in the AugMDP notation. For example, optimism is proved by Jin et al. (2021, Section 4.2, Page 9). Moreover, the regret bound is proved by Xie et al. (2023, Theorem 1). □

Therefore, Thm. D.6 shows that GOLF satisfies the oracle conditions Def. 3.1 in the AugMDP with $\varepsilon_k^{opt} = 0$ and $\mathrm{Reg}_{\mathsf{Opt}}(K) \le (H \sqrt{\mathsf{Cov}_{\mathsf{aug}} \beta K \log(K)})$.

The main question now is whether we can bound $\mathsf{Cov}_{\mathsf{aug}}$ by the original coverability $\mathsf{Cov}$, which we know is bounded for low-rank MDPs (Def. D.2) and exogenous block MDPs (Def. 3.3). We now show that if cumulative rewards live in a discrete set $\mathcal{B}$ (cf. Assump. 3.5), we can bound $\mathsf{Cov}_{\mathsf{aug}} \le |\mathcal{B}|\mathsf{Cov}$ by a simple importance sampling argument.

**Lemma D.7.** *Under Assump. 3.5, we have*

$$\mathsf{Cov}_{\mathsf{aug}} \le |\mathcal{B}|\mathsf{Cov}.$$

*Proof.*

$$\mathsf{Cov}_{\mathsf{aug}} \overset{(i)}{=} \max_{h \in [H]} \sum_{s,b,a \in \mathcal{S} \times [0,1] \times \mathcal{A}} \max_{\pi \in \Pi_{\mathsf{aug}}, b_1 \in [0,1]} d_h^{\pi, b_1}(s, b, a)$$

$$\overset{(ii)}{\le} \max_{h \in [H]} \sum_{s,b,a \in \mathcal{S} \times [0,1] \times \mathcal{A}} \max_{\pi \in \Pi_{\mathsf{aug}}, b_1 \in [0,1]} d_h^{\pi, b_1}(s, a)$$

$$\overset{(iii)}{\le} \max_{h \in [H]} \sum_{s,b,a \in \mathcal{S} \times [0,1] \times \mathcal{A}} \max_{\pi \in \Pi_{\mathsf{HD}}} d_h^{\pi}(s, a)$$

$$\overset{(iv)}{=} \max_{h \in [H]} \sum_{s,b,a \in \mathcal{S} \times [0,1] \times \mathcal{A}} \max_{\pi \in \Pi_{\mathsf{Markov}}} d_h^{\pi}(s, a)$$

$$\overset{(v)}{=} |\mathcal{B}| \max_{h \in [H]} \sum_{s,a \in \mathcal{S} \times \mathcal{A}} \max_{\pi \in \Pi_{\mathsf{Markov}}} d_h^{\pi}(s, a)$$

$$\overset{(vi)}{=} |\mathcal{B}|\mathsf{Cov},$$

where (i) is by coverability's equivalence to cumulative reachability (Xie et al., 2023, Lemma 3); (ii) is by $d^{\pi, b_1}(s, b, a) \le d^{\pi, b_1}(s, a)$ since $\mathcal{B}$ is discrete; (iii) is since $\pi, b_1$ can be viewed as a policy in the original MDP; (iv) is by the Markov optimality theorem for risk-neutral RL: $\max_{\pi \in \Pi_{\mathsf{HD}}} d_h^{\pi}(s', a')$ is equivalent to standard RL with the reward $r_h(s, a) = \mathbb{I}[(s, a) = (s', a')]$ and zero otherwise, and we know that Markovian policies are optimal for risk-neutral RL; (v) is by discreteness of $\mathcal{B}$ to collect common terms; and (vi) is again by Xie et al. (2023, Lemma 3) in the original MDP. □

We can now state our OCE regret guarantees for low-rank MDPs and exogenous block MDPs. Note these are the first risk-sensitive regret bounds for these rich-observation MDPs.

**Theorem D.8.** *In an exogenous block MDP (Def. 3.3), under Assump. 3.5, we have*

$$\mathsf{Cov}_{\mathsf{aug}} \le |\mathcal{B}||\mathcal{Z}^{\mathsf{en}}||\mathcal{A}|.$$

*Therefore, under Assump. D.5, running Alg. 1 with the GOLF oracle (cf. Alg. 5) enjoys the regret bound:*

$$\sum_{k=1}^{K} \mathrm{OCE}_u^{\star} - \mathrm{OCE}_u(\pi^k, \widehat{b}_k) \le \mathcal{O}(V_u^{\max} H \sqrt{|\mathcal{B}||\mathcal{Z}^{\mathsf{en}}||\mathcal{A}|\beta K \log(K)}).$$

*Proof.* To prove the first statement, recall that $\mathsf{Cov} \le |\mathcal{Z}^{\mathsf{en}}||\mathcal{A}|$, as was proved by Xie et al. (2023, Proposition 5). Thus, Lemma D.7 implies that $\mathsf{Cov}_{\mathsf{aug}} \le |\mathcal{B}| \cdot |\mathcal{Z}^{\mathsf{en}}||\mathcal{A}|$. To prove the second statement, simply combine our meta-algorithm guarantee in Thm. 3.2 with the GOLF oracle guarantee in Thm. D.6. □

**Theorem D.9.** *In a low-rank MDP with rank $d$ (Def. D.2), under Assump. 3.5, we have*

$$\mathsf{Cov}_{\mathsf{aug}} \le |\mathcal{B}|d|\mathcal{A}|.$$

*Therefore, under Assump. D.5, running Alg. 1 with the GOLF oracle (cf. Alg. 5) enjoys the regret bound:*

$$\sum_{k=1}^{K} \mathrm{OCE}_u^{\star} - \mathrm{OCE}_u(\pi^k, \widehat{b}_k) \le \mathcal{O}\Big(V_u^{\max} H \sqrt{|\mathcal{B}|d|\mathcal{A}|\beta K \log(K)}\Big).$$

*Proof.* To prove the first statement, recall that $\mathsf{Cov} \leq d|\mathcal{A}|$, as was proved by Huang et al. (2023, Proposition 3). Thus, Lemma D.7 implies that $\mathsf{Cov}_{\mathsf{aug}} \leq |\mathcal{B}| \cdot d|\mathcal{A}|$. To prove the second statement, simply combine our meta-algorithm guarantee in Thm. 3.2 with the GOLF oracle guarantee in Thm. D.6. $\qquad \square$

Note the above is the first risk-sensitive regret bound for low-rank MDPs. The previous Rep-UCB result ((Zhao et al., 2024) or Thm. D.4) is technically not a regret bound since the data collection policy takes uniform exploratory actions, *i.e.*, Rep-UCB can only yield PAC bounds.

### D.4. Tight bounds for CVaR via an oracle with second-order regret

Recall that Thm. 3.2 combined with UCB-VI (Thm. D.1) gave an OCE regret bound of $\mathcal{O}(V_u^{\max}\sqrt{SAHK})$. Specializing to CVaR, we have $\mathcal{O}(\tau^{-1}\sqrt{SAHK})$. A weakness in this bound is that it is not tight in $\tau$, since the minimax-optimal rate for CVaR RL actually scales with $\tau^{-1/2}$ (Wang et al., 2023a, Theorem 3.1).

In this section, we address this issue by assuming that the oracle has a second-order regret. Second-order regret, *a.k.a.* variance-dependent regret, is an instance-dependent regret bound that scales with the variance of returns throughout $K$ episodes, and is strictly stronger than the standard $\sqrt{K}$ minimax-optimal regret. Specifically, an oracle with second-order regret takes the following form.

**Assumption D.10** (Second-Order Oracle). $\mathrm{Reg}_{\mathsf{Opt}}(K) \leq \sqrt{C_1 \sum_k \mathrm{Var}(Z(\pi^k, \widehat{b}_k))} + C_2$ for constants $C_1, C_2$.

For example, optimistic model-based algorithms such as UCB-VI with Bernstein bonus (Azar et al., 2017; Zanette and Brunskill, 2019) can achieve second-order bounds in tabular MDPs. Indeed, this property of UCB-VI was used to prove the tight $\tau^{-1/2}$ rate for CVaR RL in Wang et al. (2023a); our following result generalizes this argument. Recently, distributional RL has been used to also prove second-order bounds in much more general MDPs, such as low-rank MDPs (Wang et al., 2024b; 2025a).

We also assume a continuous returns assumption of (Wang et al., 2023a).

**Assumption D.11** (Continuously Distributed Returns). For all policies $\pi \in \Pi_{\mathsf{HD}}$, the cumulative reward $Z(\pi)$ is continuously distributed with density lower-bounded by $p_{\min}$.

The following result has a sharp $\tau^{-1/2}$ dependence and generalizes the minimax-optimal results of (Wang et al., 2023a).

**Theorem D.12** ($\tau^{-1/2}$-regret for CVaR). *Under Assumps. D.10 and D.11, Alg. 1 enjoys*

$$\mathrm{Reg}_{\mathrm{CVaR}_\tau}(K) \leq 4\tau^{-1/2}\sqrt{C_1 K} + 4\tau^{-1}(C_1 p_{\min}^{-1} + C_2).$$

We note that the density lower bound $p_{\min}$ only scales a lower-order term, which is independent of $K$. The proof uses the fact that the AugMDP return variance is at most $\tau$ under the true initial budget $b_k^\star = \arg\max_{b \in [0,1]}\{b + V_{\mathsf{aug}}^{\pi^k}(s_1, b)\}$. Then, under Assump. D.11, the approximation $(\widehat{b}_k - b_k^\star)^2$ can be related to $\mathrm{Reg}_{\mathsf{Opt}}$, which leads to a self-bounding inequality that solves to $\mathrm{Reg}_{\mathsf{Opt}}(K) \leq \sqrt{C_1 K\tau} + 2C_1 p_{\min}^{-1} + 2C_2$. Compared to Wang et al. (2023a, Theorem 5.5), our result is more general (applies beyond tabular MDPs) and Thm. D.12 also sharpens lower-order terms ($p_{\min}$ is not multiplied with $K^{1/4}$).

*Proof.* First, we want to show: under continuous returns (Assump. D.11), we have

$$\mathrm{Reg}_{\mathsf{Opt}}(K) \leq 2\sqrt{C_1 K\tau} + 2C_1 p_{\min}^{-1} + 2C_2. \tag{10}$$

Recall for CVaR, the normalized reward in the AugMDP is precisely precisely $(\widehat{b}_k - Z(\pi^k, \widehat{b}_k))_+$. Let $b_k^\star$ denote the true

$\tau$-quantile of $Z(\pi^k, \widehat{b}_k)$.

$$
\begin{aligned}
\text{Reg}_{\text{Opt}}(K) & \\
&\overset{(i)}{\leq} \sqrt{C_1 \sum_k \text{Var}(b_k^\star - Z(\pi^k, \widehat{b}_k)_+)} + \sqrt{C_1 \sum_k \text{Var}((\widehat{b}_k - Z(\pi^k, \widehat{b}_k))_+ - (b_k^\star - Z(\pi^k, \widehat{b}_k))_+)} + C_2 \\
&\overset{(ii)}{\leq} \sqrt{C_1 K \tau} + \sqrt{C_1 \sum_k \text{Var}((\widehat{b}_k - Z(\pi^k, \widehat{b}_k))_+ - (b_k^\star - Z(\pi^k, \widehat{b}_k))_+)} + C_2 \\
&\overset{(iii)}{\leq} \sqrt{C_1 K \tau} + \sqrt{C_1 \sum_k (\widehat{b}_k - b_k^\star)^2} + C_2 \\
&\overset{(iv)}{\leq} \sqrt{C_1 K \tau} + \sqrt{2 C_1 p_{\min}^{-1} \text{Reg}_{\text{Opt}}(K)} + C_2 \\
&\leq \sqrt{C_1 K \tau} + C_1 p_{\min}^{-1} + \tfrac{1}{2} \text{Reg}_{\text{Opt}}(K) + C_2. \hspace{2cm} \text{(AM-GM)}
\end{aligned}
$$

(i) holds since $\sqrt{\text{Var}(X + Y)} \leq \sqrt{\text{Var}(X)} + \sqrt{\text{Var}(Y)}$ for any random variables $X, Y$. (ii) holds since $\text{Var}(X_+) \leq \mathbb{E}[X^2 \mathbb{I}[X \geq 0]] \leq \Pr(X \geq 0)$ for any bounded $X \in [-1, 1]$ and $b_k^\star$ is the true $\tau$-quantile. (iii) holds since $(\widehat{b}_k - Z_k)_+ - (b_k^\star - Z_k)_+ \leq |\widehat{b}_k - b_k^\star|$ almost surely. (iv) holds by (Wang et al., 2023a, Lemma G.10): it states that under Assump. D.11, the choice of $\widehat{b}_k$ ensures:

$$
(\widehat{b}_k - b_k^\star)^2 \leq 2 p_{\min}^{-1} (\widehat{V}_{1,k}(s_1, \widehat{b}_k) - V_{\text{aug}}^{\pi^k, 1}(s_1, \widehat{b}_k)).
$$

Rearranging $\text{Reg}_{\text{Opt}}(K)$ implies the desired Eq. (10). Therefore, we can conclude the proof applying Thm. 3.2 and using $V_u^{\max} = \tau^{-1}$ and Eq. (10):

$$
\text{Reg}_{\text{OCE}}(K) = 4 \tau^{-1/2} \sqrt{C_1 K} + 4 \tau^{-1} (C_1 p_{\min}^{-1} + C_2).
$$

$\square$

## E. Proofs for Policy Optimization Meta-Algorithm

We begin by proving a stronger global convergence guarantee than the one stated in the main paper.

**Theorem E.1** (Strong Global Convergence). *Under Assump. 3.5 and Def. 4.1, we have*

$$
\sum_{k=1}^K \text{OCE}_u^\star - \text{RLB}^{(k)} \leq |\mathcal{B}| V_u^{\max} \text{Reg}_{\text{PO}}(K).
$$

*Proof.*

$$
\begin{aligned}
\sum_{k=1}^K \text{OCE}_u^\star - \text{RLB}^{(k)} &= \sum_{k=1}^K \{ b_1^\star + V_{\text{aug}}^\star(s_1, b_1^\star) \} - \max_{b \in [0,1]} \left\{ b + V_{\text{aug}}^{\pi^k, 1}(s_1, b) \right\} \\
&\leq \sum_{k=1}^K V_{\text{aug}}^\star(s_1, b_1^\star) - V_{\text{aug}}^{\pi^k, 1}(s_1, b_1^\star) \\
&\leq \sum_{k=1}^K |\mathcal{B}| \tfrac{1}{|\mathcal{B}|} \sum_{b \in \mathcal{B}} \left( V_{\text{aug}}^\star(s_1, b) - V_{\text{aug}}^{\pi^k, 1}(s_1, b) \right) \\
&\leq |\mathcal{B}| \text{Reg}_{\text{PO}}(K),
\end{aligned}
$$

where the last step uses the global convergence premise of POALG in the AugMDP. $\square$

This is stronger than the global convergence in the main paper (cf. Thm. 4.2), since the RLB approximately lower bounds $\text{OCE}_u(Z(\pi^k, \widehat{b}_k))$, as we prove next. Therefore, Thm. 4.2 is a direct consequence of Thm. E.1 and the following RLB lemma (Eq. (7)).

**Lemma 4.3** (RLB). *The $\text{RLB}^{(k)}$ approximately lower bounds the true OCE of $\pi^k$ with initial budget $\widehat{b}_k$:*

$$
\text{RLB}^{(k)} - \text{OCE}_u(\pi^k, \widehat{b}_k) \leq 2 |\mathcal{B}| \varepsilon_k^{\text{po}}. \tag{7}
$$

*Moreover, the lower bound is tight on average:*

$$
\sum_k \text{OCE}_u(\pi^k, \widehat{b}_k) - \text{RLB}^{(k)} \leq |\mathcal{B}| V_u^{\max} \text{Reg}_{\text{PO}}(K).
$$

*Proof.* First, we prove Eq. (7). By importance sampling and the value estimate guarantee of Def. 4.1: for all $b_1 \in \mathcal{B}$, we have $|V_1^{\pi^k}(s_1, b_1) - \widehat{V}_1^{\pi^k}(s_1, b_1)| \leq |\mathcal{B}|\varepsilon_k^{\text{po}} =: \varepsilon_k'$. Thus,

$$
\begin{aligned}
\text{OCE}_u(\pi^k, \widehat{b}_k) &= \max_{b \in \mathcal{B}}\{b + \mathbb{E}[u(Z(\pi^k, \widehat{b}_k) - b)]\} \\
&\geq \{\widehat{b}_k + \mathbb{E}[u(Z(\pi^k, \widehat{b}_k) - \widehat{b}_k)]\} && (\widehat{b}_k \in \mathcal{B}, \text{ by Eq. (5)}) \\
&= \{\widehat{b}_k + V_{\text{aug}}^{\pi^k, 1}(s_1, \widehat{b}_k)\} && (\text{def. of } V^{\pi^k}) \\
&\overset{(i)}{\geq} \{\widehat{b}_k + \widehat{V}_1^{\pi^k}(s_1, \widehat{b}_k)\} - \varepsilon_k' \\
&= \max_{b_1 \in \mathcal{B}}\{b_1 + \widehat{V}_1^{\pi^k}(s_1, b_1)\} - \varepsilon_k' && (\text{def. of } \widehat{b}_k, \text{ by Eq. (5)}) \\
&\overset{(ii)}{\geq} \max_{b_1 \in \mathcal{B}}\{b_1 + V_{\text{aug}}^{\pi^k, 1}(s_1, b_1)\} - 2\varepsilon_k' \\
&= \text{RLB}^{(k)} - 2\varepsilon_k', && (\text{def. of RLB, by Eq. (6)})
\end{aligned}
$$

where the inequalities (i,ii) are due to the value estimate guarantee. This finishes the proof of Eq. (7).

To prove the second statement, simply apply the stronger global convergence result Thm. E.1 with the fact that $\text{OCE}_u(Z(\pi^k, \widehat{b}_k)) \leq \text{OCE}_u^\star$. $\qquad\square$

**Theorem 4.4** (Local Improvement). *Under Assump. 3.5 and assuming* POALG *satisfies the approximate improvement criterion of Def. 4.1, running Alg. 2 ensures that:*

$$\forall k \in [K] : \text{RLB}^{(k+1)} \geq \text{RLB}^{(k)} - |\mathcal{B}|\varepsilon_k^{\text{po}}.$$

*Proof.* Let $b_k^\star = \arg\max_{b \in \mathcal{B}}\{b + V_{\text{aug}}^{\pi^k, 1}(s_1, b)\}$. Then,

$$
\begin{aligned}
\text{RLB}^{(k)} - \text{RLB}^{(k+1)} &= \{b_k^\star + V_{\text{aug}}^{\pi^k, 1}(s_1, b_k^\star)\} - \max_{b \in [0,1]}\{b + V_{\text{aug}}^{\pi^{k+1}, 1}(s_1, b)\} \\
&\leq V_{\text{aug}}^{\pi^k, 1}(s_1, b_k^\star) - V_{\text{aug}}^{\pi^{k+1}, 1}(s_1, b_k^\star) \\
&\leq |\mathcal{B}|\frac{1}{|\mathcal{B}|}\sum_{b \in \mathcal{B}}(V_{\text{aug}}^{\pi^k, 1}(s_1, b) - V_{\text{aug}}^{\pi^{k+1}, 1}(s_1, b)) \\
&\leq |\mathcal{B}|\varepsilon_k^{\text{po}},
\end{aligned}
$$

where the last step uses local improvement premise of POALG in the AugMDP. $\qquad\square$

### E.1. Primer on Natural Policy Gradient in Finite-Horizon MDPs

In this section, we provide a primer on natural policy gradient (NPG) for finite-horizon MDPs and its guarantees, supplementing the case study (Sec. 4.1) of the main paper. Our analysis is based on the infinite-horizon analysis of Agarwal et al. (2021). Let $d_1$ denote an fixed initial state distribution. For a policy $\pi$, let $Q_h^\pi(s_h, a_h) = \mathbb{E}_\pi[r_h + r_{h+1} + \cdots + r_H \mid s_h, a_h]$ denote the $Q$-function, $V_h^\pi(s_h) = \mathbb{E}_{a_h \sim \pi(s_h)}[Q_h^\pi(s_h, a_h)]$ denote the value function, and $A_h^\pi(s, a) = Q_h^\pi(s, a) - V_h^\pi(s)$ denote the advantage function.

We consider policies $\pi^\theta = (\pi_1^\theta, \ldots, \pi_H^\theta)$ parameterized by weight vectors $\theta = (\theta_1, \ldots, \theta_H)$ where $\theta_h \in \mathbb{R}^m$. Initialize $\theta_h^{(0)} = \mathbf{0}$ for all $h \in [H]$. Consider the generic update rule for $k = 1, 2, \ldots, K$: for all $h$,

$$\theta_h^{(k+1)} = \theta_h^{(k)} + \eta w_h^{(k)},$$

where $w_h^{(k)} \in \mathbb{R}^m$ have $\ell_2$-norm at most $W$. Let $\pi^k$ denote the policy parameterized by $\theta^{(k)}$. Then, we have the following lemma that bounds the sub-optimality of the update rule.

**Lemma E.2.** *Suppose the log-probability parameterization $\theta_h \mapsto \log \pi_{\theta_h}(a \mid s)$ is $\beta$-smooth for all $h, s, a$. Then, setting $\eta = \sqrt{\frac{2\log(A)}{\beta K W^2}}$, we have*

$$\sum_{k=1}^K \mathbb{E}_{s_1 \sim d_1}[V_1^\star(s_1) - V_1^{\pi^k}(s_1)] \leq HW\sqrt{2\beta K \log(A)} + \sum_{h,k} \text{err}_{h,k},$$

*where* $\text{err}_{h,k} := \mathbb{E}_{s_1 \sim d_1, \pi^\star}[A_h^{\pi^k}(s_h, a_h) - w_h^{(k)} \cdot \nabla_{\theta_h} \log \pi_{\theta_h^{(k)}}(a_h \mid s_h)].$

*Proof.* First, notice that the smoothness assumption implies that for all $h, s, a$,

$$\log \pi_h^{k+1}(a \mid s) - \log \pi_h^k(a \mid s) \geq (\theta_h^{(k+1)} - \theta_h^{(k)}) \cdot \nabla_{\theta_h} \log \pi_{\theta_h^{(k)}}(a \mid s) - \frac{\beta}{2} \|\theta_h^{(k+1)} - \theta_h^{(k)}\|_2^2$$

$$\geq \eta w_h^{(k)} \cdot \nabla_{\theta_h} \log \pi_{\theta_h^{(k)}}(a \mid s) - \frac{\beta \eta^2 W^2}{2}. \qquad \text{(by update rule)}$$

Rearranging, we have

$$w_h^{(k)} \cdot \nabla_{\theta_h} \log \pi_{\theta_h^{(k)}}(a \mid s) \leq \eta^{-1} \log(\pi_h^{k+1}(a \mid s)/\pi_h^k(a \mid s)) + \frac{\beta \eta W^2}{2}. \qquad (11)$$

Then, we can bound the sub-optimality at the $k$-th round by:

$$\mathbb{E}_{s_1 \sim d_1}[V_1^\star(s_1) - V_1^{\pi^k}(s_1)]$$
$$= \sum_{h=1}^H \mathbb{E}_{s_1 \sim d_1, \pi^\star}[A_h^{\pi^k}(s_h, a_h)] \qquad \text{(by performance difference lemma)}$$
$$= \sum_h \mathrm{err}_{h,k} + \mathbb{E}_{s_1 \sim d_1, \pi^\star}[w_h^{(k)} \cdot \nabla_{\theta_h} \log \pi_{\theta_h^{(k)}}(a_h \mid s_h)]$$
$$\leq \sum_h \mathrm{err}_{h,k} + \eta^{-1} \mathbb{E}_{s_1 \sim d_1, \pi^\star}[\log(\pi_h^{k+1}(a_h \mid s_h)/\pi_h^k(a_h \mid s_h))] + \frac{\beta \eta W^2}{2} \qquad \text{(by Eq. (11))}$$
$$= \sum_h \mathrm{err}_{h,k} + \eta^{-1} \mathbb{E}_{s_1 \sim d_1, \pi^\star}[D_{KL}(\pi_h^\star(s_h) \parallel \pi_h^k(s_h)) - D_{KL}(\pi_h^\star(s_h) \parallel \pi_h^{k+1}(s_h))] + \frac{\beta \eta W^2}{2},$$

where $D_{KL}$ is the KL-divergence.

Finally, summing over $k$ implies the final result by telescoping:

$$\sum_{k=1}^K \mathbb{E}_{s_1 \sim d_1}[V_1^\star(s_1) - V_1^{\pi^k}(s_1)]$$
$$\leq \frac{\beta \eta H W^2 K}{2} + \eta^{-1} \sum_h \mathbb{E}_{s_1 \sim d_1, \pi^\star}[D_{KL}(\pi_h^\star(s_h) \parallel \pi_h^1(s_h)) - D_{KL}(\pi_h^\star(s_h) \parallel \pi_h^K(s_h))] + \sum_{h,k} \mathrm{err}_{h,k}$$
$$\leq \frac{\beta \eta H W^2 K}{2} + \eta^{-1} H \log(A) + \sum_{h,k} \mathrm{err}_{h,k} \qquad (\pi^{(1)} \text{ is uniform})$$
$$= 2\sqrt{\beta H^2 W^2 K \log(A)/2} + \sum_{h,k} \mathrm{err}_{h,k}. \qquad \text{(choice of } \eta)$$

This finishes the proof. $\qquad \square$

A natural idea is to set $w_h^{(k)}$ to minimize the $\mathrm{err}_{h,k}$ error terms, which exactly motivates the NPG update. Specifically, the (idealized) NPG update vector is defined as:

$$\widetilde{w}_h^{(k)} = \arg\min_{\|w\|_2 \leq W} L_h(w; \theta^{(k)}, d_h^{\pi^k}),$$
$$\text{where } L_h(w_h; \theta', \nu_h) := \mathbb{E}_{s_h, a_h \sim \nu_h}[(A_h^{\pi_{\theta'}}(s_h, a_h) - w_h \cdot \nabla \log \pi_{\theta_h'}(a_h \mid s_h))^2],$$

for $\theta_h \in \mathbb{R}^d$ and $\nu_h \in \Delta(\mathcal{S} \times \mathcal{A})$. In other words, the idealized NPG update vector minimizes the squared error term while having Euclidean norm at most $W$. Since the true mean is unknown, we approximate the above using samples. Thus, the actual NPG update we consider minimizes the empirical error

$$w_h^{(k)} := \arg\min_{\|w\|_2 \leq W} \widehat{L}_h(w; \theta^{(k)}, \{s_{h,i}, a_{h,i}\}_{i \in [N]}), \qquad (12)$$

where $\widehat{L}_h(w_h; \theta', \{s_{h,i}, a_{h,i}\}_{i \in [N]}) = \frac{1}{N} \sum_i (A_h^{\pi_{\theta'}}(s_{h,i}, a_{h,i}) - w_h \cdot \nabla \log \pi_{\theta_h'}(a_{h,i} \mid s_{h,i}))^2$ and $s_{h,i}, a_{h,i} \sim d_h^{\pi^k}$ are $N$ *i.i.d.* samples from roll-outs.

We now bound the sub-optimality of the NPG update. We define two sources of error. First, there is the statistical estimation error from using the empirical loss $\widehat{L}_h$ instead of the true $L_h$. This error is expected to converge at a $O(1/\sqrt{N})$ rate or faster, where $N$ is the number of samples per round. Second, there is a transfer / approximation error that measures the performance of the best population vector $\widetilde{w}_h^{(k)}$ under the optimal policy's visitations $d_h^{\pi^\star}$. This quantity is small if the training policies for data collection cover the traces of the optimal policy $\pi^\star$. The first estimation error is also measured under the distribution induced by $\pi^\star$, but can be handled by using the relative condition number, which is a weaker measure of coverage than $\ell_\infty$ density ratio bounds.

The following results are based on the infinite-horizon MDP analysis of Agarwal et al. (2021, Section 6.3).

**Assumption E.3.** The update vectors generated by NPG satisfy for all $k \in [K], h \in [H]$:

1. (Excess risk) The statistical estimation error is bounded by $\varepsilon_{\text{stat}}$:

$$L(w_h^{(k)}; \theta^{(k)}, d_h^{\pi^k}) - L(\widetilde{w}_h^{(k)}; \theta^{(k)}, d_h^{\pi^k}) \leq \varepsilon_{\text{stat}}.$$

2. (Coverage) The relative condition number of the covariance under $d_h^{\pi^k}$ and $d_h^{\pi^\star}$ is bounded by $\kappa$:

$$\sup_{w \in \mathbb{R}^d} \frac{w^\top \Sigma_{h,k,\pi^\star} w}{w^\top \Sigma_{h,k,\pi^k} w} \leq \kappa,$$

where $\Sigma_{h,k,\pi'} := \mathbb{E}_{\pi'}[\nabla_{\theta_h} \log \pi_h^k(a_h \mid s_h)(\nabla_{\theta_h} \log \pi_h^k(a_h \mid s_h))^\top]$.

3. (Transfer Error) Assume the transfer error is bounded by $\varepsilon_{\text{bias}}$:

$$L(\widetilde{w}^{(k)}; \theta^{(k)}, d_h^{\pi^\star}) \leq \varepsilon_{\text{bias}}.$$

**Theorem E.4** (Agnostic NPG). *Under Assump. E.3, running the NPG update in Eq. (12) enjoys:*

$$\sum_{k=1}^{K} \mathbb{E}_{s_1 \sim d_1}[V_1^\star(s_1) - V_1^{\pi^k}(s_1)] \leq HW\sqrt{2\beta K \log(A)} + HK\sqrt{\varepsilon_{bias}} + HK\sqrt{\kappa \varepsilon_{stat}}.$$

This is a finite-horizon analog of Agarwal et al. (2021, Theorem 6.2), where their effective horizon $1/(1-\gamma)$ is replaced with the horizon $H$.

*Proof.* The proof focuses on decomposing the error term in Lemma E.2.

$$\text{err}_{h,k} = \mathbb{E}_{s_1 \sim d_1, \pi^\star}[A_h^{\pi^k}(s_h, a_h) - w_h^{(k)} \cdot \nabla_{\theta_h} \log \pi_{\theta_h^{(k)}}(a_h \mid s_h)]$$

$$= \underbrace{\mathbb{E}_{s_1 \sim d_1, \pi^\star}[A_h^{\pi^k}(s_h, a_h) - \widetilde{w}_h^{(k)} \cdot \nabla_{\theta_h} \log \pi_{\theta_h^{(k)}}(a_h \mid s_h)]}_{approx.error} + \underbrace{(\widetilde{w}_h^{(k)} - w_h^{(k)}) \cdot \mathbb{E}_{s_1 \sim d_1, \pi^\star}[\nabla_\theta \log \pi_{\theta_h}(a_h \mid s_h)]}_{est.error}.$$

Let $d_h^\star = d_{1,d_1}^{\pi^\star}$ be the $h$-th visitation distribution of rolling in $\pi^\star$ from $d_1$. The approximation error can be handled by:

$$\mathbb{E}_{s_1 \sim d_1, \pi^\star}[A_h^{\pi^k}(s_h, a_h) - \widetilde{w}_h^{(k)} \cdot \nabla_{\theta_h} \log \pi_{\theta_h^{(k)}}(a_h \mid s_h)]$$

$$\leq \sqrt{\mathbb{E}_{s_1 \sim d_1, \pi^\star}[(A_h^{\pi^k}(s_h, a_h) - \widetilde{w}_h^{(k)} \cdot \nabla_{\theta_h} \log \pi_{\theta_h^{(k)}}(a_h \mid s_h))^2]} = \sqrt{L_h(\widetilde{w}_h^{(k)}; \theta^{(k)}, d_h^\star)}.$$

The estimation error can be handled by:

$$\left| (\widetilde{w}_h^{(k)} - w_h^{(k)}) \cdot \mathbb{E}_{s_1 \sim d_1, \pi^\star}[\nabla_{\theta_h} \log \pi_{\theta_h^{(k)}}(a_h \mid s_h)] \right| \leq \|\widetilde{w}_h^{(k)} - w_h^{(k)}\|_{\Sigma_{h,k,\pi^\star}}.$$

Then by definition of the relative condition number $\kappa_{h,k} = \|(\Sigma_{h,k,\pi^{(k)}})^{-1/2}\Sigma_{h,k,\pi^\star}(\Sigma_{h,k,\pi^{(k)}})^{-1/2}\|_2$, we have

$$\|\widetilde{w}_h^{(k)} - w_h^{(k)}\|_{\Sigma_{h,k,\pi^\star}} \leq \sqrt{\kappa_{h,k}}\|\widetilde{w}_h^{(k)} - w_h^{(k)}\|_{\Sigma_{h,k,\pi^k}}$$

$$= \sqrt{\kappa_{h,k}(L_h(w_h^{(k)}, \theta^{(k)}, d_h^{\pi^k}) - L_h(\widetilde{w}_h^{(k)}, \theta^{(k)}, d_h^{\pi^k}))}.$$

Therefore, we have shown that

$$\text{err}_{h,k} \leq \sqrt{L_h(\widetilde{w}_h^{(k)}; \theta^{(k)}, d_h^\star)} + \sqrt{\kappa_{h,k}(L_h(w_h^{(k)}; \theta^{(k)}, d_h^{\pi^k}) - L_h(\widetilde{w}_h^{(k)}; \theta^{(k)}, d_h^{\pi^k}))}$$

$$\leq \sqrt{\varepsilon_{\text{bias}}} + \sqrt{\kappa \varepsilon_{\text{stat}}},$$

which concludes the proof. $\qquad \square$

