# OpenReview forum: "A Reductions Approach to Risk-Sensitive Reinforcement Learning with Optimized Certainty Equivalents"
_ICML.cc/2025/Conference — ICML 2025 poster_

### Official Review · Reviewer_74co · 2025-03-13

**Overall Recommendation:** 4

**Summary:**

The paper propose two meta-algorithms for solving static risk sensitive MDPs by using the augmented formulation for OCE objectives. The first one uses models based oracles and the other one using policy optimization based oracles. Regret bounds are analyzed with the assumption that the oracles are reasonably well designed.

**Claims And Evidence:**

The claims are clearly presented with theoretical results (regret bounds) and some computational results.

**Essential References Not Discussed:**

The related works are relatively complete.

**Experimental Designs Or Analyses:**

The experiments are reasonable and interesting to see.

**Methods And Evaluation Criteria:**

The methods and evaluations are suitable.

**Other Comments Or Suggestions:**

I am a little confused by the notion on the value function with all the sub/sup scripts. What's the relationship between: $ \hat V_{1,k} $, $ V_1^{\pi^k} $, $ V_{\text{aug}}^{\pi^k} $?

**Other Strengths And Weaknesses:**

The paper is well written. The introduction is clear and relationship with prior works are well discussed.
The augmented approach is practical due to the objective being intuitive. The presented meta-algorithms cover a wide range of application and problem setups.
The biggest weakness is that the results are hardly surprising as the augmented MDP approach is well known and has been studied extensively. Plus given a known oracle with assumption/definition 3.1 & 4.1, all regret bounds seem to be easy to derive.

**Questions For Authors:**

Please see above.

**Relation To Broader Scientific Literature:**

RSRL is known to be hard to solve. This paper provides a good summary for existing methods for static OCE objectives, and demonstrated efficient implementations are possible.

**Theoretical Claims:**

Proofs in the main section are checked.

---

> ### Author Rebuttal · Authors · 2025-03-25
>
> Dear Reviewer 74co,
>
> Thank you for your encouraging review and strong support! We truly appreciate the time and effort that have been invested in providing constructive comments.
>
> Regarding your question about notation on the value function:
> * $\hat V_{1,k}$ is an estimated augmented value function from the Optimistic Oracle (Def 3.1). The subscript $1,k$ denotes it's at step $h=1$ and is from $k$-th round of the meta-algorithm (Alg 1).
> * $V^{\pi^k}\_1$ and $V^{\pi^k}\_{\text{aug}}$ are both denoting the true augmented value function of policy $\pi^k$ at step $h=1$. Thanks for catching this duplicated notation, we will consolidate these two notations into one.
>
> Regarding your concern about the novelty of our work:
> While the AugMDP has been studied for specific risk measures like CVaR, our work makes several novel contributions beyond straightforward application of this approach:
>
> 1. We establish the first risk-sensitive PAC bounds for exogenous block MDPs (Thm 3.6 in Sec 3.1) – not captured by low-rank MDPs (Zhao et al. 2024) and requires novel techniques to handle the interplay between coverability and the augmented MDP.
> 2. We derive the first risk-sensitive bounds for policy-gradient algorithms and prove a novel local improvement property in risk-sensitive RL (Thms 4.2 and 4.4 in Sec 4). This bridges the gap in the literature where policy gradient methods lacked theoretical guarantees in risk-sensitive settings.
> 3. By abstracting out oracles, we provide a unifying framework for studying the large class of OCE risk measures that were previously disconnected in the risk-sensitive RL literature (e.g., Bastani et al. 2022, Wang et al. 2023, Zhao et al. 2024). Our framework not only simplifies analysis but also enables new algorithmic design (leading to the above two contributions), applicable across multiple risk measures. Our Related Works section (Sec 1.1) provides a more comprehensive discussion that positions our novel contributions within the RL literature. We are happy to answer any questions on our technical contributions.
>
> Please let us know if you have any other questions!

---

### Official Review · Reviewer_t1og · 2025-03-14

**Overall Recommendation:** 3

**Summary:**

In this paper, the authors study the risk-sensitive RL problem to optimize a risk-measure of cumulative rewards. A family of risks called Optimized Certainty Equivalents (OCEs) are considered, and this includes popular risk measures such as CVaR and entropic risk. Two algorithms have been proposed in the paper, based on optimism principle and policy gradient, respectively. By forming an augmented Markov Decision Process (MDP), the formalism developed for risk-neutral MDP has been utilized in the paper. In the first algorithm, bounds that generalize prior results in CVaR RL, has been established. For the policy gradient based algorithm, monotone improvement and global convergence guarantees have been established under some assumptions. Empirically, it has been verified that the proposed algorithms learn the optimal history-dependent policy and Markovian policies fail to achieve optimality.

## update after rebuttal
 The response addresses my major questions. I have slightly raised my score to 3 (as the generalization of AugMDP to the OCE setting does not seem to be very novel and is highly dependent on Rep-UCB in the literature).

**Claims And Evidence:**

Yes

**Essential References Not Discussed:**

No

**Experimental Designs Or Analyses:**

Experimental designs seem to be fine.

**Methods And Evaluation Criteria:**

Yes

**Other Comments Or Suggestions:**

Not applicable

**Other Strengths And Weaknesses:**

The paper is well-written, however difficult to follow at some places. It introduces many concepts but some of the claims are largely based on existing works. There are some assumptions that requires further justifications.

**Questions For Authors:**

1.	Why do we need to assume that $Z(\pi)\in[0,1]$? Since this is a summation of rewards, this may create a dependence between per-stage reward and horizon length. What happens when this condition is violated? This assumption seems far from reality.
2.	The augmented MDP for OCE has a state space which has a much higher dimensionality than the original state space. The approach for construction for augmented MDP does not seem to be very efficient.
3.	The optimistic oracle outputs an optimistic value function if two conditions, viz., optimism and bounded regret, are satisfied. How do one guarantee that these conditions will be satisfied? Also, second condition states that the regret needs to be sublinear, i.e., $O(\sqrt(K))$? Is to possible to achieve logarithmic regret in the second condition? What will be the implication of the result on the overall convergence?
4.	More intuition needs to be provided while defining $V_u^{\max}$. Please quantify this in details.
5.	The basic concepts behind exogenous block MDP needs to be elaborated.
6.	Theorem 4.2 guarantees global convergence if POA_LG has small value estimation error. How do one guarantees this small value estimation error?
7.	Overall, the paper introduces many notions towards a single objective. This impacts the readability of the paper, requiring more intuitions to be provided throughout the paper. Novelty of the proposed methodologies are limited, e.g., the generalization of AugMDP to the OCE setting does not seem to be very novel and is highly dependent on Rep-UCB in the literature.

**Relation To Broader Scientific Literature:**

A family of risks called Optimized Certainty Equivalents (OCEs) are considered, and this includes popular risk measures such as CVaR and entropic risk. Two algorithms have been proposed in the paper, based on optimism principle and policy gradient, respectively. In the first algorithm, bounds that generalize prior results in CVaR RL, has been established. For the policy gradient based algorithm, monotone improvement and global convergence guarantees have been established under some assumptions.

**Theoretical Claims:**

The proofs seem to be okay.

---

> ### Author Rebuttal · Authors · 2025-03-25
>
> Dear Reviewer t1og,
>
> Thanks very much for providing helpful feedback. We truly appreciate the time and effort that have been invested in providing constructive comments. Please find our responses below.
>
> 1. Our assumption to normalize cumulative rewards is actually *more general* than normalizing per-step reward, as it allows for reward values that drastically vary across actions and states. For example, the reward sequence $0, 0, ..., 0, H$ is allowed under our framework (after scaling our normalization & bounds by $H$). This is in contrast to assuming that per-step rewards are in $[0,1]$ and cumulative rewards are in $[0,H]$, the arguably more popular assumption in RL theory, which only allows for dense rewards and precludes the above sparse reward example. A helpful resource that helped us understand the benefit of normalizing cumulative rewards is the paper http://proceedings.mlr.press/v75/jiang18a/jiang18a.pdf (which we cite in line 145) and their COLT 2018 talk https://youtu.be/If63ZSpEiSs at the 5:10 mark.
> 2. We note that the AugMDP for OCE only has one extra dimensionality compared to the original MDP, since the augmented state $b_h$ is a scalar. Moreover, the AugMDP is very easy to simulate, since $b_h$ has a known and deterministic transition function $b_{h+1}=b_h-r_h$, which adds essentially no overhead to the original MDP. Please see "Remark: the AugMDP is easy to simulate" in Lines 184-191 for more details.
> 3. Thanks for the question. Regarding how to satisfy the two conditions (optimism and bounded regret) in Def 3.1, we note these conditions are satisfied by many optimistic algorithms in RL even when applied to the AugMDP. For example, UCB-VI satisfies these conditions in tabular MDPs, Rep-UCB in low-rank MDPs and GOLF in MDPs with exogenous block MDPs. We refer the reviewer to the paragraphs after Thm 3.2 and the text in 3.1 for several ways to satisfy these two conditions.
> Regarding logarithmic regret in the second condition, this is certainly captured by our framework; the second condition only states that regret should be sub-linear, which certainly includes $O(\log K)$ regret as well. The implication on the OCE convergence is stated in Thm 3.2, which shows that the OCE regret is bounded by the oracle's regret multiplied by a constant $V^{\max}_u$ that doesn't change the rate in $K$. Thus, an oracle with logarithmic regret would yield an OCE algorithm with logarithmic regret.
> 4. Thanks for the suggestion, we'll certainly add more intuition about $V^{\max}_u$: this constant intuitively captures is the statistical complexity of learning the OCE with utility $u$; for example, it is $1/\tau$ for CVaR at level $\tau$ and this is intuitive because smaller levels of $\tau$ measures a smaller sub-population and thus requires more samples to learn. We will list and discuss more examples of $V^{\max}_u$ in the main text (they are currently in Table 4 of the appendix due to page limit).
> 5. We will also elaborate more about the exogenous block MDP in the revision.
> 6. Thanks for the question. In theory, small value estimation error can be ensured by online supervised learning or regression, which is fairly standard (Agarwal et al. 2021). In practice, policy optimization algorithms such as REINFORCE with value baseline or PPO maintain a separate value network and interleave in gradient updates to the value network by minimizing squared loss (Schulman et al. 2017). We will add more discussions in the revision.
> 7. While we concur with the reviewer that our approaches are inspired by existing algorithms, we believe our work nonetheless contributes several novel and insightful results to risk-sensitive RL:
>
>     1. We establish the first risk-sensitive PAC bounds for exogenous block MDPs (Thm 3.6 in Sec 3.1) – not captured by low-rank MDPs (Zhao et al. 2024) and requires novel techniques to handle the interplay between coverability and the augmented MDP.
>     2. We derive the first risk-sensitive bounds for policy-gradient algorithms and prove a novel local improvement property in risk-sensitive RL (Thms 4.2 and 4.4 in Sec 4). This bridges the gap in the literature where policy gradient methods lacked theoretical guarantees in risk-sensitive settings.
>     3. By abstracting out oracles, we provide a unifying framework for studying the large class of OCE risk measures that were previously disconnected in the risk-sensitive RL literature (e.g., Bastani et al. 2022, Wang et al. 2023, Zhao et al. 2024). Our framework not only simplifies analysis but also enables new algorithmic design (leading to the above two contributions), applicable across multiple risk measures. Our Related Works section (Sec 1.1) provides a more comprehensive discussion that positions our novel contributions within the RL literature. We are happy to answer any questions on our technical contributions.
>
> Please let us know if you have any other questions!

---

> > ### Comment · Reviewer_t1og · 2025-04-05
> >
> > I thank the authors for their detailed response. However, responses towards questions 3 and 6 need more clarity. While authors have provided some example algorithms which satisfy these conditions (optimism, bounded regret and small value estimation error), it is not clear how these conditions will be satisfied. More intuitions need to be provided regarding these.

---

> > > ### Author Response · Authors · 2025-04-06
> > >
> > > Dear Reviewer t1og,
> > >
> > > Thanks so much for your follow-up comments. We truly appreciate the opportunity to elaborate on our responses to questions 3 and 6.
> > >
> > > **Q3: How to satisfy optimism & bounded regret in Def 3.1**
> > >
> > > In the following, we describe separately how these two conditions are satisfied by model-based and model-free oracles in the augmented MDP:
> > >
> > > * Model-based oracles estimate the MDP's transition kernel $P_h$. For example, optimistic model-based oracles include UCB-VI for tabular MDPs and Rep-UCB for low-rank MDPs. To learn $P_h(s'\mid s,a)$, UCB-VI computes the maximum likelihood estimate by counting visitations of $(s,a,s')$, which is feasible since state and action spaces are finite in tabular MDPs. Rep-UCB generalizes this idea to low-rank MDPs by maximizing log-likelihood with a linear function class. Importantly, these methods only need to estimate transitions for the original states $s_h$, and *not* for the augmented budgets $b_h$, because the budget's transition is a known function: $b_{h+1}=b_h-r_h$. *This means that our approach introduces no extra statistical or computational complexity for learning transitions in the augmented MDP.*
> > > Both algorithms compute optimistic functions $\hat Q_{h,k},\hat V_{h,k}$ by planning in the learned model with an exploration bonus, which is large enough to ensure optimism but small enough to ensure sub-linear regret when the bonuses are summed across episodes. In sum, UCB-VI and Rep-UCB satisfy optimism and bounded regret via the same standard argument from, respectively, (Azar et al., 2017) and (Uehara et al., 2022), since learning the transition in the AugMDP doesn't add extra complexity.
> > > * Model-free oracles directly learn $Q,V$ without first estimating the transition kernel. The model-free algorithm we focus on is GOLF, which maintains a version space $\mathcal{F}\_k$ that contains functions with nearly-minimal Bellman error (across steps $h$) on the aggregated data at iteration $k$. The threshold for "nearly-minimal" is set at $\beta=\Theta(\log(KH|\mathcal{F}|/\delta))$ scale so that two properties hold under Bellman completeness (a standard condition for model-free algs that we also posit): (1) the optimal $Q^{\star}\_{aug}$ is an element of $\mathcal{F}\_k$, and (2) the total Bellman residuals over $K$ rounds is $O(\sqrt{K})$. To ensure optimism, GOLF selects the element in $\mathcal{F}\_k$ with the maximum value at $h=1$, which is optimistic by the above Property (1). Moreover, performance difference lemma and Property (2) can be used to bound the regret by $O(\sqrt{K})$. Specifically, we invoke the result from (Xie et al., 2023) which bounds GOLF's regret by $\tilde{O}(H\sqrt{\text{Cov}\cdot K\log(|\mathcal{F}|/\delta)})$, where $\text{Cov}$ is the coverability the MDP; notably, the coverability is small in the challenging exogenous block MDP (Ex-BMDP). Since GOLF is applied in the AugMDP, we then bound the AugMDP's coverability by the original MDP's coverability in Lemma D.7. Putting this together, we conclude that GOLF has bounded regret in augmented MDPs with low coverability, including the challenging Ex-BMDP; we note this is the first risk-sensitive bound for Ex-BMDPs.
> > >
> > >
> > > **Q6: How to satisfy small value estimation error in Def 4.1**
> > >
> > > This condition can be satisfied by reducing to standard policy evaluation, which is well-studied in RL. A simple approach is on-policy evaluation: rollout $\pi^k$ for each initial state to obtain an unbiased estimate of the value, which can be used for Monte Carlo estimation or regression. In risk-sensitive settings, we may however want to avoid rollouts for policy evaluation and instead apply off-policy evaluation (OPE) using the available data from prior rounds $1, 2, ..., k-1$, i.e. replay buffer in practice. A simple and practical OPE alg is fitted-Q evaluation (FQE) which simply minimizes the policy TD error. Under Bellman completeness, FQE's estimation error is bounded by $\tilde{O}(\sqrt{\frac{C^{\pi^k}\log(1/\delta)}{n}})$, where $C^{\pi^k}$ is the density ratio between $\pi^k$'s visitation distribution and the data distribution [1,2]. There are also more sophisticated OPE methods that relax assumptions and obtain faster convergence [3,4]. In conclusion, multiple established methods in policy evaluation can effectively satisfy the small value estimation error in Def 4.1.
> > >
> > > Thanks again for your time and effort in providing valuable questions and comments! We'll certainly address and discuss all of the above in our revision.
> > >
> > > **Citations for Policy Evaluation**
> > >
> > > [1] Wang et al, "A Fine-grained Analysis of Fitted Q-evaluation: Beyond Parametric Models", ICML 2024.
> > >
> > > [2] Chang et al, "Learning Bellman complete representations for offline policy evaluation", ICML 2022.
> > >
> > > [3] Kallus and Uehara, "Double Reinforcement Learning for Efficient Off-Policy Evaluation in Markov Decision Processes", ICML 2020.
> > >
> > > [4] Yang et al, "Off-Policy Evaluation via the Regularized Lagrangian", NeurIPS 2020.

---

### Official Review · Reviewer_ZPJ1 · 2025-03-19

**Overall Recommendation:** 2

**Summary:**

This work studies risk-sensitive reinforcement learning, where the target is to maximize $\max_{\pi} \max_{b} \\{b+E_{\pi}[u(\sum_{h=1}^H r_h - b)]\\}$ where $u$ is some utility function. For this problem, the authors prove that by augmentation, there exists a Markovian policy which reaches the optimality. Based on UCB-VI, the authors designed an algorithm with provable regret bound. Then authors combine their formulation with several other frameworks to generalize the convergence rate.

**Claims And Evidence:**

The theoretical claims are well supported by the regret bounds, and the empirical results are too minor to support the claim that the augmentation method is powerful.

**Essential References Not Discussed:**

The related works are well discussed.

**Experimental Designs Or Analyses:**

Only a few experiments were conducted on a toy example, which does not significantly influence the evaluation of this work.

**Methods And Evaluation Criteria:**

NA

**Other Comments Or Suggestions:**

I do not have any other comments.

**Other Strengths And Weaknesses:**

Strength:
The authors  designed a  meta-algorithm which could be applied to many different OCE-RL settings,  including the challenging exogenous block MDP.

Weakness:

My major concern is that the method in this work is very limited in both technical novelty and conceptual insight. Firstly, by the definition of OCE, it is very nature to formulate the problem with an augmented state space, which is roughly a combination of traditional RL (to learn $P$) with dynamic programming (to learn $b$). Secondly, the proofs in this work are merely straightforward applications of existing algorithms, which significantly limits the technical novelty.

**Questions For Authors:**

1. Why do you consider the assumption that $Z(\pi)\in [0,1]$, instead of the classical assumption that each $r_h \in [0,1]$?

**Relation To Broader Scientific Literature:**

This work focuses on risk-sensitive RL with OCE reward, which provides insights for exploring the risk-sensitive RL problem.

**Theoretical Claims:**

As far as I see, the proofs are correct.

---

> ### Author Rebuttal · Authors · 2025-03-25
>
> Dear Reviewer ZPJ1,
>
> Thanks very much for providing helpful feedback. We truly appreciate the time and effort that have been invested in providing constructive comments. Please find our responses below.
>
> **Reviewer: The empirical results are too minor to support the claim that the augmentation method is powerful.**
>
> **Authors' reply:** Yes, we concur that our empirical results are only a proof-of-concept, while our main contribution is theoretical. Our empirical simulation is specifically to show that, in a minimal MDP, our methods indeed learn the optimal risk-sensitive policy while previous algorithms with Markov policies have bounded performance (i.e. they fail to learn the optimal policy), which motivates further research to step away from Markov policies and to use the AugMDP.
>
> **Reviewer: My major concern is that the method in this work is very limited in both technical novelty and conceptual insight. Firstly, by the definition of OCE, it is very nature to formulate the problem with an augmented state space, which is roughly a combination of traditional RL (to learn $P$) with dynamic programming (to learn $b$). Secondly, the proofs in this work are merely straightforward applications of existing algorithms, which significantly limits the technical novelty.**
>
> **Authors' reply:** While we concur with the reviewer that our approaches are inspired by existing algorithms, we believe our work nonetheless contributes several novel and insightful results to risk-sensitive RL:
> 1. We establish the first risk-sensitive PAC bounds for exogenous block MDPs (Thm 3.6 in Sec 3.1) – not captured by low-rank MDPs (Zhao et al. 2024) and requires novel techniques to handle the interplay between coverability and the augmented MDP.
> 2. We derive the first risk-sensitive bounds for policy-gradient algorithms and prove a novel local improvement property in risk-sensitive RL (Thms 4.2 and 4.4 in Sec 4). This bridges the gap in the literature where policy gradient methods lacked theoretical guarantees in risk-sensitive settings.
> 3. By abstracting out oracles, we provide a unifying framework for studying the large class of OCE risk measures that were previously disconnected in the risk-sensitive RL literature (e.g., Bastani et al. 2022, Wang et al. 2023, Zhao et al. 2024). Our framework not only simplifies analysis but also enables new algorithmic design (leading to the above two contributions), applicable across multiple risk measures. Our Related Works section (Sec 1.1) provides a more comprehensive discussion that positions our novel contributions within the RL literature.
>
> **Reviewer: Why do you consider the assumption that $Z(\pi)\in[0,1]$, instead of the classical assumption that each $r_h\in[0,1]$?**
>
> **Authors' reply:** Our assumption to normalize cumulative rewards is actually *more general* than normalizing per-step reward as it allows for reward values that drastically vary across actions and states. For example, the reward sequence $0, 0, ..., 0, H$ is allowed under our framework (after scaling our normalization by $H$). This is in contrast to the arguably more popular assumption in the RL theory community that per-step rewards are in $[0,1]$ and cumulative rewards are in $[0,H]$, which only allows for dense rewards and precludes the above sparse reward example. A helpful resource that helped us understand the benefit of normalizing cumulative rewards is the paper http://proceedings.mlr.press/v75/jiang18a/jiang18a.pdf (which we cite at line 145) and their COLT 2018 talk https://youtu.be/If63ZSpEiSs at the 5:10 mark.
>
> Please let us know if you have any other questions!

---

### Official Review · Reviewer_5fhe · 2025-03-20

**Overall Recommendation:** 3

**Summary:**

**I am very unfamiliar with this topic. I will maintain the lowest confidence level.**

This paper develops a study on risk-sensitive RL, which is formulated through OCE. Two meta algorithms are proposed with further analysis.

**Claims And Evidence:**

The paper propose an augmented MDP for the OCE problem, which bypasses the challenges. The theorems indeed support the claims that the augmentations can help solve OCE.

**Essential References Not Discussed:**

n/a

**Experimental Designs Or Analyses:**

The experiment only contains one simple MDP, although neural policies and advanced algorithms like PPO are considered.

**Methods And Evaluation Criteria:**

The experiments seem to only contain a simple MDP, which could benefit from more evaluations.

**Other Comments Or Suggestions:**

n/a

**Other Strengths And Weaknesses:**

The paper could benefit from more explanation. For example, in eq (1) defining OCE, it is unclear to me what this definition means. Some examples could help.

In def 3.1, an oracle is introduced. Is this oracle used in later algorithm designs? And how do you implement it?

**Questions For Authors:**

See above.

**Relation To Broader Scientific Literature:**

n/a

**Theoretical Claims:**

I checked the proof of THM 2.1. Although the proof is not very easy to follow, it seems to be correct.

---

> ### Author Rebuttal · Authors · 2025-03-25
>
> Dear Reviewer 5fhe,
>
> Thanks for providing your helpful feedback. We truly appreciate the time and effort that have been invested in providing constructive comments. Please find our responses below.
>
> **Reviewer: The paper could benefit from more explanation. For example, in eq (1) defining OCE, it is unclear to me what this definition means. Some examples could help.**
>
> **Authors' reply:** Thanks for the suggestion. Eq (1) is the risk-sensitive RL objective defined using the static OCE risk measure and is the key objective we aim to optimize in the paper. Right after Eq (1), we provide a few examples of OCE risk measures, such as CVaR when u is the hinge utility and Markowitz's mean-variance when u is the quadratic utility. We will elaborate more on these examples in the revision by moving some content from Appendix B to the main text.
>
> **Reviewer: In def 3.1, an oracle is introduced. Is this oracle used in later algorithm designs? And how do you implement it?**
>
> **Authors' reply:** Thanks for the question! Yes, the optimistic oracle in Def 3.1 is used in the meta-algorithm Alg 1, and similarly the policy gradient oracle in Def 4.1 is used in the meta-algorithm Alg 2. In our paper, we discuss several examples of how to implement these oracles for different settings. For example, for the optimistic oracle in Def 3.1, we discuss three ways to implement it using (1) UCB-VI, (2) Rep-UCB and (3) GOLF. Moreover, for our policy gradient oracle in Def 4.1, we discuss natural policy gradients (NPG) as an example instantiation. By employing different oracles, we are able to derive guarantees in several types of MDPs, e.g. UCB-VI is used in tabular MDPs, Rep-UCB in low-rank MDPs, GOLF in exogenous block MDPs, NPG in MDPs with good initial state distribution, where the latter two are novel to risk-sensitive RL. If you have any further questions about how these oracles are implemented, we would be happy to answer them!

---

### Official Review · Reviewer_1jGh · 2025-03-20

**Overall Recommendation:** 4

**Summary:**

This paper studies risk-sensitive reinforcement learning (RSRL) with the goal of learning a history-dependent policy that optimizes Optimized Certainty Equivalents (OCE) risk measures of cumulative rewards. The authors propose two meta-algorithms, one based on optimism and another on policy gradients, that reduce the RSRL problem to risk-neutral RL in an augmented Markov Decision Process (MDP). Theoretical guarantees, including regret bounds and convergence properties, are established, and empirical results demonstrate the effectiveness of the proposed methods in learning optimal history-dependent policies.

**Claims And Evidence:**

The claims are supported by rigorous mathematical proofs and experiments.

**Essential References Not Discussed:**

See weakness part.

**Experimental Designs Or Analyses:**

I have checked the soundness of the experimental design.

**Methods And Evaluation Criteria:**

The methods and evaluation criteria are appropriate.

**Other Comments Or Suggestions:**

I recommend that the authors delve deeper into the computational efficiency of the proposed algorithms. For instance, they could explore and discuss what the major challenges are in developing a computationally and statistically efficient algorithm for OCE RL.


If my concerns and comments are adequately addressed, I will consider raising my score.

**Other Strengths And Weaknesses:**

### Strength
Strengths include the originality of the reductions approach to OCE RL, the significance of providing the first risk-sensitive bounds for exogenous block MDPs, and the clarity of the theoretical and empirical results.

### Weakness
My main concern is that the author has not referenced the studies on the Lipschitz risk measure in [1, 2]. Although not previously discussed in related works, I believe that the OCE is encompassed within the Lipschitz risk measure. The factor $V_{\max}^u$ closely resembles the Lipschitz constant described in [1, 2], which might give a formal way to determine this factor. Since [2] has already proposed an optimism - based algorithm grounded in distribution learning for the Lipschitz risk, I suggest that the author offer a comprehensive comparison with [2]. This would help to clearly elucidate the novelty of the first contribution of this paper.

[1] Liang, H., & Luo, Z. (2024, April). Regret bounds for risk-sensitive reinforcement learning with lipschitz dynamic risk measures. In International Conference on Artificial Intelligence and Statistics (pp. 1774-1782). PMLR.

[2] Chen, Y., Zhang, X., Wang, S., & Huang, L. (2024, July). Provable Risk-Sensitive Distributional Reinforcement Learning with General Function Approximation. In International Conference on Machine Learning (pp. 7748-7791). PMLR.

**Questions For Authors:**

Given that discretizing the total reward in Risk-Sensitive Reinforcement Learning (RSRL) is a prevalent technique, as demonstrated in [3, 4], discretization appears to be a rational approach to address the discrete reward assumption (Assump. 3.5). Will this method works in OCE RL?

[3] Bastani, O., Ma, J. Y., Shen, E., & Xu, W. (2022). Regret bounds for risk-sensitive reinforcement learning. Advances in Neural Information Processing Systems, 35, 36259-36269.

[4] Chen, Y., Du, Y., Hu, P., Wang, S., Wu, D., & Huang, L. Provably Efficient Iterated CVaR Reinforcement Learning with Function Approximation and Human Feedback. In The Twelfth International Conference on Learning Representations.

**Relation To Broader Scientific Literature:**

n/a

**Theoretical Claims:**

I have checked the correctness of the main theoretical claims. The proofs appear to be logically sound and technically correct. However, the constant $V_{\max}^u$ is not defined formally, which is confusing.

---

> ### Author Rebuttal · Authors · 2025-03-25
>
> Dear Reviewer 1jGh,
>
> Thanks very much for providing helpful feedback. We truly appreciate the time and effort that have been invested in providing constructive comments. Please find our responses below.
>
> **Reviewer: the constant $V^{\max}_u$ is not defined formally**
>
> **Authors' reply:** We kindly refer the reviewer to line 130 where $V^{\max}_u$ is defined. We will make sure to highlight this definition in the revision with examples and connection to Lipschitz risk measure (see next reply).
>
> **Reviewer: My main concern is that the author has not referenced the studies on the Lipschitz risk measure in [1, 2]. Although not previously discussed in related works, I believe that the OCE is encompassed within the Lipschitz risk measure. The factor $V^{\max}_u$ closely resembles the Lipschitz constant described in [1, 2], which might give a formal way to determine this factor. Since [2] has already proposed an optimism - based algorithm grounded in distribution learning for the Lipschitz risk, I suggest that the author offer a comprehensive comparison with [2]. This would help to clearly elucidate the novelty of the first contribution of this paper.**
>
> **Authors' reply:** Thanks very much for pointing out Lipschitz risk measures. You're right that they indeed encompass the OCE with Lipschitz constant at most $V^{\max}_u$ and we'll cite and discuss this as another way to interpret $V^{\max}_u$. Our work is distinguished from prior Lipschitz risk RL [1,2] as follows:
> * [1] studies RL with dynamic risk objective (a.k.a. iterated risk) $\rho(r_1+\rho(r_2+\rho(r_3+...)))$ which measures per-step risk and is different from our static risk objective $\rho(r_1+r_2+r_3+...)$ which measures trajectory-wise risk. Both objectives are important and we discuss dynamic risks in Lines 110-128 of Related Works, to which we'll add a citation to [1].
> * [2] derives regret bounds for static Lipschitz risks under function approximation which is more related to our setting. However, a key technical assumption made in [2] is bounded Bellman eluder dimension or witness rank, which are insufficient to capture the challenging exogenous block MDP setting. Indeed, the Bellman eluder dimension can grow with the size of the exogenous state space which is exponentially large or infinite (Xie et al. 2023). In contrast, we employ a coverability argument in the augmented MDP to obtain the first PAC bound for exogenous block MDP in risk-sensitive RL. Thus, our first contribution is distinct from [2] since our framework can deal with the challenging exogenous block MDP while [2] cannot. Moreover, our work includes other contributions such as studying policy gradient methods that are complementary to optimistic algorithms.
>
> **Reviewer: I recommend that the authors delve deeper into the computational efficiency of the proposed algorithms. For instance, they could explore and discuss what the major challenges are in developing a computationally and statistically efficient algorithm for OCE RL.**
>
> **Authors' reply:** Thanks for this suggestion, we'll certainly add more discussion in the revision. In short, our work shows that the computational and statistical complexity of OCE RL algorithms can largely be reduced to standard risk-neutral RL in the AugMDP. In terms of computation, the main costs come from 1) optimizing over $b$, which is efficient under reward discretization, and 2) querying the underlying risk-neutral RL oracle. In terms of sample efficiency, our main theorems (Thm 3.2 & 4.2) ensure that the OCE regret is bounded by the underlying oracle's regret up to constant factors. Thus, in both cases, the major challenge is devising a risk-neutral oracle that is efficient within the AugMDP. We will make sure to incorporate these discussions in the revision.
>
> **Reviewer: Given that discretizing the total reward in Risk-Sensitive Reinforcement Learning (RSRL) is a prevalent technique, as demonstrated in [3, 4], discretization appears to be a rational approach to address the discrete reward assumption (Assump. 3.5). Will this method works in OCE RL?**
>
> **Authors' reply:** Thanks for this question! Yes, discretizing rewards as in [3, 4] can indeed by used to address the discrete reward assumption (Assump 3.5), at the cost of a slower regret rate. Specifically, using a discretization width of $\epsilon$ introduces additional regret of $O(K\epsilon)$ and creates $|\mathcal{B}|=1/\epsilon$ atoms. Thus, our regret bound in Thm 3.6 would become $O(\sqrt{K/\epsilon} + K\epsilon)$. The $\epsilon$ which minimizes this is $\epsilon = \Theta(1/K^{1/3})$, which leads to a regret bound of $O(K^{2/3})$ which is sub-linear and meaningful. In other words, we can remove Assump 3.5 by discretizing rewards with bin width of $1/K^{1/3}$, and our regret bound in Thm 3.6 becomes $O(V^{\max}_uHK^{2/3} \sqrt{ Z^{en}A\log(|\mathcal{F}|/\delta) })$. We'll make sure to discuss this in the revision.
>
> Please let us know if you have any other questions!

---

> > ### Comment · Reviewer_1jGh · 2025-04-04
> >
> > I appreciate the response, as it addresses most of my concerns. I have raised my score to 4.

---

### Decision · Program_Chairs · 2025-05-01

**Decision:**

Accept (poster)

**Comment:**

The reviewers agree that the paper describes a useful algorithm for a practical setting. The writing is clear, and the paper is organized well. Most of the results are supported by proofs, although some of the proofs (such as Thm 2.1 and Thm 3.2) are much more just proof sketches than actual proofs. Although the proofs are not rigorous, most of the results are plausible. The numerical results are limited but illustrate the setting and algorithm quite well.

Although the results in the paper are novel, they are somewhat incremental. The extension from CVaR to OCE is relatively straightforward, and not developed in great depth. The reviewers did not find much methodological innovation in the paper. It is not clear if there are any technical difficulties that need to be overcome.